# Compressed Decentralized Learning with Error-Feedback under Data Heterogeneity

## Abstract

Decentralized learning distributes the training process across multiple nodes, enabling collaborative model training without relying on a central server. Each node performs local training using its own data, with model updates exchanged directly between connected nodes within a given network topology. Various algorithms have been developed within this decentralized learning framework and have been proven to converge under specific assumptions. However, two key challenges remain: 1) ensuring robust performance with both a high degree of gradient compression and data heterogeneity, and 2) providing a general convergence upper bound under commonly used assumptions. To address these challenges, we propose the *Discounted Error-Feedback Decentralized Parallel Stochastic Gradient Descent (DEFD-PSGD)* algorithm, which efficiently manages both high levels of gradient compression and data heterogeneity, without sacrificing communication efficiency. The core idea is to introduce controllable residual error feedback that effectively balances the impact of gradient compression and data heterogeneity. Additionally, we develop novel proof techniques to derive a convergence upper bound under relaxed assumptions. Finally, we present experimental results demonstrating that DEFD-PSGD outperforms other state-of-the-art decentralized learning algorithms, particularly in scenarios involving high compression and significant data heterogeneity.

## 1 Introduction

In recent years, decentralized learning has become an important technology in machine learning due to its computational scalability for parallel computing (Nedic & Ozdaglar, 2009; Lian et al., 2017), communication efficiency (Tang et al., 2019; 2018; Koloskova et al., 2019; Pu & Nedić, 2021), and data locality for data privacy (Wangni et al., 2018; Reisizadeh et al., 2019). Specifically, decentralized learning distributes the training process across multiple nodes in a network and allows them to collaboratively train a shared model without a central server. We consider the scenario where each node possesses its own data. During training, every node performs local updates based on its own data, where different nodes may have different data distributions. Then, each node exchanges model updates with connected nodes according to the network's connectivity.

**General Challenges.** In the decentralized learning framework, one of the most persistent challenges is the communication efficiency. To address this, recent research has focused on designing algorithms that use compressed model updates (Wangni et al., 2018; Reisizadeh et al., 2019; Tang et al., 2019; 2018; Koloskova et al., 2019). One of their drawbacks is the degradation of performance when the degree of compression level is high or when the compression is biased. To mitigate the performance degradation due to compression, *error feedback* can be applied. Its core idea is to accumulate and correct the errors that occur during gradient updates, thereby improving the accuracy and stability of the learning process. Earlier works on error feedback focuses on centralized settings with parameter server architecture (Stich et al., 2018; Wu et al., 2018; Karimireddy et al., 2019). However, because the local models of nodes are not fully synchronized in the decentralized setting, directly applying error feedback to the seminal DCD-PSGD algorithm (Tang et al., 2018) can be ineffective. Koloskova et al. (2019) introduced the CHOCO-PSGD algorithm where error feedback is applied to the decentralized learning procedure. By using error feedback in the decentralized learning scenario, both theoretical and empirical studies show that CHOCO-PSGD is robust to gradient compression. However, one drawback of CHOCO-PSGD is its inefficiency in handling the scenario

of non-IID data across nodes, especially when the degree of data heterogeneity is high. Hence, in decentralized learning, it is challenging to handle both the high degree of gradient compression and high degree of data heterogeneity simultaneously.

**Motivating Example.** To illustrate the aforementioned challenges, we consider a network of 20 nodes, each connected to 4 neighbors. As shown in Figure 1, applying error feedback directly to DCD-PSGD leads to divergence. Additionally, in this example, when the data is highly heterogeneous (e.g., with Dirichlet parameter $\alpha = 0.05$, see later for definitions), CHOCO-PSGD also fails due to its specific model update mechanism, despite handling data compression well under homogeneous local data distributions. Moreover, the DCD-PSGD algorithm without error feedback, although convergent, suffers from significant performance degradation in this scenario due to the high compression (e.g., top-$k$ compression with $k = 0.1$, see later for definitions). Therefore, managing both a high degree of compression and significant data heterogeneity simultaneously is challenging.

In this paper, we address the following important question:

*Is there a decentralized learning algorithm that can handle both high degree of gradient compression and high degree of data heterogeneity with a provable performance guarantee?*

There are two main challenges in answering this question. *First*, as discussed before, applying naive error feedback in the decentralized setting can cause divergence, as shown in Figure 1. This is because different nodes can have different local model parameters in decentralized learning. With error feedback, the model updates computed on such different local models can be partially accumulated locally and transmitted to neighboring nodes at a later time. In addition, each node can only transmit its updates to its neighbors, which means that it can take a long time for an update transmitted by each node to reach nodes that are many hops away from

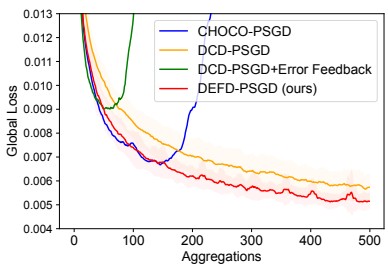

Figure 1: Global loss for FashionMNIST dataset, using Dirichlet parameter $\alpha = 0.05$ and top-$k$ with $k = 0.1$. The network consists of 20 nodes and each connected 4 nodes.

it. This can cause a high degree of asynchronicity in different nodes' model parameters, which may ultimately cause divergence of the learning process. In Koloskova et al. (2019), the authors mitigate this potential negative impact of error feedback by reducing the weights of neighboring models during model aggregation. However, this leads to very limited information exchange among neighboring nodes, making CHOCO-PSGD ineffective in handling high data heterogeneity. Hence, we need to design a new error feedback algorithm capable of handling both high compression ratios and significant data heterogeneity. *Second*, analyzing the convergence of decentralized learning with error feedback is challenging. Existing literature either lacks convergence analysis or relies on strict assumptions, such as bounded gradients and consistent gradient averages across iterations. Therefore, we need to develop new proof techniques to handle error feedback under relaxed assumptions.

To overcome the above challenges, we propose the *Discounted Error-Feedback Decentralized Parallel Stochastic Gradient Descent (DEFD-PSGD)* algorithm, where we develop a novel approach to incorporate the error feedback to DCD-PSGD. Our DEFD-PSGD algorithm includes a new discount factor multiplied to the error feedback term to control its impact on the convergence behavior. We show that DEFD-PSGD can address both obstacles effectively and have the same communication efficiency with the state-of-the-art algorithms. In particular, we demonstrate that DEFD-PSGD outperforms DCD-PSGD in scenarios of high data compression and surpasses CHOCO-PSGD when the data distribution is highly heterogeneous. This can be seen from our motivating example in Figure 1. In addition, we introduce new proof techniques to analyze the convergence upper bound of the proposed DEFD-PSGD with error feedback under relaxed assumptions.

**Our Contributions.** In this paper, our novel contributions are summarized as follows.

1. We propose the DEFD-PSGD algorithm, designed to effectively manage both model compression and data heterogeneity. This algorithm innovatively incorporates residual error feedback into the decentralized learning process and introduces a new discount parameter, $\gamma$, to regulate the impact of error feedback during the decentralized training.

2. We analyze the convergence upper bound of DEFD-PSGD using a novel approach to handle error feedback. Under the most commonly used assumptions in decentralized learning, the

derived convergence upper bound achieves the same order of convergence rate as existing literature, but under significantly relaxed parameter conditions, such as allowing a broader range for the model compression ratio. The insights of the convergence results are also discussed.

3. The proposed DEFD-PSGD algorithm outperforms state-of-the-art algorithms, particularly in scenarios involving high degree of model compression and highly heterogeneous data distributions.

## 2 RELATED WORKS

Under a centralized framework, Stich et al. (2018); Wu et al. (2018) introduce error-compensated SGD with compressed updates, including analysis for (strongly) convex functions. Karimireddy et al. (2019) analyze the impact of error feedback for Sign-SGD and other compression schemes for weakly convex and non-convex functions. However, the centralized error feedback algorithms require information aggregation among all nodes.

In decentralized learning, gossip algorithms are among the most popular approaches due to their simplicity (Kempe et al., 2003; Johansson & Johansson, 2008; Boyd et al., 2006; Lian et al., 2017). In a gossip algorithm, each node first computes the Stochastic Gradient Descent (SGD) locally and then sends its update to its connected nodes. Under certain network connectivity assumptions, Lian et al. (2017) show that the convergence rate of this algorithm is $\mathcal{O}(1/\sqrt{nT})$, where $n$ is the total number of nodes and $T$ is the total number of iterations. Notably, this convergence rate matches that of centralized SGD, where a central server aggregates the SGDs and updates the model parameters in each iteration.

To reduce communication overhead between connected nodes, compression techniques such as quantization and sparsification are applied on the model updates transmitted among nodes. These techniques are widely used in the federated learning literature (Basu et al., 2019; Haddadpour et al., 2021; Alistarh et al., 2017; Bernstein et al., 2018; Shlezinger et al., 2020; Reisizadeh et al., 2020; Wangni et al., 2018; Sattler et al., 2019; Albasyoni et al., 2020; Gorbunov et al., 2021; Alistarh et al., 2018; Stich & Karimireddy, 2020; Karimireddy et al., 2019) as well as decentralized learning literature (Wangni et al., 2018; Reisizadeh et al., 2019; Tang et al., 2019; 2018; Koloskova et al., 2019). In particular, Tang et al. (2018) introduce the compression methods on the model updates shared in gossip algorithms and propose the DCD-PSGD and ECD-PSGD algorithms, demonstrating that their convergence when using unbiased compressors or compressors with bounded noise. However, directly applying these compression techniques can adversely affect the convergence rate of gossip algorithms, and may even lead to divergence if the compression noise is large. The reason is that the lost information due to compression is not compensated.

The CHOCO-PSGD algorithm, introduced by Koloskova et al. (2019), is another approach aimed at improving the compression rate in decentralized learning. It incorporates a consensus step size parameter to regulate model aggregation at each node along with error feedback. However, it struggles to handle highly heterogeneous data distributions effectively. Recently, a few other algorithms were designed based on CHOCO-PSGD. In particular, AdaG-PSGD (Aketi et al., 2024) was proposed to dynamically adjust the consensus steps used in Koloskova et al. (2019). Choudhary et al. (2024) proposed Q-SADDLe algorithm, which applied additional gradient descent to seek flatter loss landscapes in decentralized setting. This flatter loss landscapes allow more compression to alleviate the local over-fitting with non-IID data. Aketi et al. (2021) proposed Sparse-Push (SP) algorithm, which includes additional communication round to handle the time-varying network topologies. Nassif et al. (2024) proposed the DEF-ATC algorithm, introducing a damping coefficient in front of the updates in Koloskova et al. (2019). DEF-ATC promises a maintaining performance while reducing communication overheads in small step-size regime. Since these algorithms share the same limitation as CHOCO-PSGD when dealing with high degree of data heterogeneity, in this paper, we use DCD-PSGD and CHOCO-PSGD as our baselines in the main paper. The comparison among all the algorithms will be presented in the Appendix A.5.

In this paper, the proposed DEFD-PSGD algorithm incorporates residual error feedback to effectively manage biased compressors and those with higher compression noise. We validate its effectiveness both analytically and empirically. In addition, we introduces a new discount parameter, $\gamma$,

in DEFD-PSGD. We empirically study the impact of $\gamma$ and show that $\gamma$ can effectively balance the impact of gradient compression and data heterogeneity.

# 3 PROBLEM FORMULATION

**Objective Function.** Let the total number of nodes be $n$, we consider a decentralized optimization problem as follows:

$$\min_{x \in R^N} \ f(x) := \frac{1}{n} \sum_{i=1}^{n} f_i(x), \tag{1}$$

where $x \in \mathbb{R}^N$ represents the model parameters and $f_i(x) = \mathbb{E}_{\xi \sim D_i} \left[ F_i(x; \xi) \right], i \in \{1, 2, \ldots, n\}$. Here, $F_i(x; \xi)$ represents the local loss function of node $i$, $D_i$ denotes local data distribution of node $i$ and $\xi$ is node $i$'s data samples from the local data distribution $D_i$.

**Decentralized Setting.** In decentralized learning algorithms, such as Decentralized Parallel Stochastic Gradient Descent (D-PSGD), a real connectivity matrix $W$ is employed, which restricts each node to communicate only with its connected nodes. After sending the corresponding local model to its connected nodes, each node $i$ computes a weighted average update according to the real connectivity matrix $W$, specifically as $x^i = \sum_{j=1}^{n} W_{ij} x^j$, where $W_{ij}$ represents weight of node $j$ related to node $i$, and it is non-negative. Specifically, $W_{ij}$ is positive if nodes $i$ and $j$ are connected, and $W_{ij}$ is equal to zero if nodes $i$ and $j$ are not connected. We make some assumptions for the real connectivity matrix $W$ in Assumption 1 that is presented later.

**Decentralized Parallel Stochastic Gradient Descent.** Before discussing our proposed algorithm, we first introduce the D-PSGD algorithm, which typically involves the following steps and notations.

At the beginning of each iteration, each node computes the averaged model parameters according to the update rule $x_t^i = \sum_{j=1}^{n} W_{ij} x_t^j$. Simultaneously, each node $i$ randomly selects the samples $\xi_t^i$ according to the local data distribution $D_i$, then uses the samples $\xi_t^i$ and the current model $x_t^i$ to compute the local stochastic gradient $\nabla F_i \left( x_t^i; \xi_t^i \right)$. The model parameter for the next iteration is updated according to the update rule $x_{t+1}^i = x_t^i - \eta \nabla F_i \left( x_t^i; \xi_t^i \right)$, where $\eta$ denotes the learning rate.

To simplify the notations, we define the vector $X_t := [x_t^1, x_t^2, \ldots, x_t^n]$ and $G(X_t) := [G_1(X_t^1), G_2(X_t^2), \ldots, G_n(X_t^n)] := [\nabla F_1(x_t^1, \xi_t^1), \nabla F_2(x_t^2, \xi_t^2), \ldots, \nabla F_n(x_t^n, \xi_t^n)]$ for each iteration $t$. Therefore, the general update rule for D-PSGD can be written as $X_{t+1} = X_t - \eta G(X_t)$.

# 4 DEFD-PSGD ALGORITHM

In this section, we introduce and discuss the details of the proposed DEFD-PSGD algorithm.

## 4.1 PROPOSED ALGORITHM

We design the proposed DEFD-PSGD algorithm based on the general decentralized parallel gradient descent algorithm. The key idea of DEFD-PSGD is to introduce controllable residual error feedback on the local model update, using a control parameter $\gamma$, to balance the impacts of gradient compression and data heterogeneity. The DEFD-PSGD algorithm is described in Algorithm 1 and consists of two main stages.

**Computation and Communication.** The first stage involves local computation and updates aggregation, including the steps in Lines 2 to 8 of Algorithm 1. Specifically, in Lines 3 and 4, each node $i$ randomly selects the samples $\xi_t^i$ according to local data distribution $D_i$ at the beginning of each iteration $t$ and computes the local stochastic gradients $\nabla F_i(x_t^i; \xi_t^i)$. Node $i$ then computes temporary model parameters $x_{t+\frac{1}{2}}^i$ using the weighted average model $\sum_{j=1}^{n} w_{ij} x_t^{i,j}$ and the local stochastic gradients $\nabla F_i(x_t^i; \xi_t^i)$. Here, the $x_t^i$ denotes the local model parameters on node $i$, and $x_t^{i,j}$ represents the model parameters of connected node $j$ stored at node $i$. This setup allows node $i$ to access the model parameters of connected nodes without requiring additional communication rounds. In Line 5, we compute the update vector $b_t^i$. We incorporate the residual error $e_t^i$ from the past iterations, scaled by a hyperparameter called discount parameter $\gamma$, which adjusts the impact

---

**Algorithm 1:** DEFD-PSGD Algorithm

**Input:** $\eta > 0, W, T$
**Output:** $\{x_t^i\}$
**Initialize:** $e_0^i \leftarrow 0, \forall i, \{x_0^i\}_{i=1}^n = x_0, \{x_0^{i,j}\}_{j=1}^n = x_0$

1 **for** $t \leftarrow 0, 1, 2, ..., T-1$ **do**
2   **for** *each node* $i \leftarrow 1, 2, ..., n$ **do**
3     Randomly sample $\xi_t^i$ from local dataset;
4     $x_{t+\frac{1}{2}}^i \leftarrow \sum_{j=1}^n w_{ij} x_t^{i,j} - \eta \nabla F_i(x_t^i; \xi_t^i)$
5     $b_t^i \leftarrow x_{t+\frac{1}{2}}^i - x_t^i + \gamma e_t^i$;
6     $v_t^i \leftarrow C_t^i(b_t^i)$;
7     $e_{t+1}^i \leftarrow b_t^i - v_t^i$;
8     Send $v_t^i$ and receive $v_t^j$;

9 **for** *each node* $i \leftarrow 1, 2, ..., n$ **do**
10   Update local parameters:
11   $x_{t+1}^i \leftarrow x_t^i + v_t^i$
12   Update neighbor's parameters:
13   **for** *each* $j, w_{ij} \neq 0$ **do**
14     $x_{t+1}^{i,j} \leftarrow x_t^{i,j} + v_t^j$

---

of the residual error $e_t^i$. In Line 6, we apply the compressor $C(\cdot)$ to obtain the transmitted update vector $v_t^i$, thereby reducing the communication load. In Line 7, we accumulate the error from this iteration for future computations. Finally, in Line 8, each node $i$ transmits the update vector $v_t^i$ to its connected nodes $j \in \{1, 2, \ldots, n\}$ according to the real connectivity matrix $W$.

**Model Update.** The second stage is the model updating process, which is described in Lines 9 to 14 of Algorithm 1. In Line 11, node $i$ updates the local model parameters following the update rule $x_{t+1}^i = x_t^i + v_t^i$. In Line 14, node $i$ updates the model parameters of its connected nodes $j \in \{1, 2, \ldots, n\}$ using the update vector received from node $j$ during the communication process in Line 8. We would like to emphasize that in Line 11 and 14, the update rule for local model of node $i$ and its neighbors' models are identical by adding the compressed vectors $v_t^i$ or $v_t^j$. Therefore, each node's model is synchronized with its neightbors' model.

## 4.2 CONVERGENCE ANALYSIS

Before we show the convergence upper bound of Algorithm 1, we introduce some assumptions which are commonly used in the literature (Lian et al., 2017; Tang et al., 2018; Koloskova et al., 2019).

**Assumption 1.** *Throughout this paper, we assume:*

- *Symmetric doubly stochastic matrix:* The real connectivity matrix $W$ is symmetric, satisfying $W = W^\top$, and doubly stochastic, satisfying and $W\mathbf{1}_n = \mathbf{1}_n$, where $\mathbf{1}_n$ is a vector of all one entries with length $n$.

- *Spectral gap:* Let $\lambda_i(W)$ denote the $i$-th largest eigenvalue of $W$. Then, given $W$, we define $\rho := \max\{|\lambda_2(W)|, |\lambda_n(W)|\}$ and assume $\rho < 1$.

- *Lipschitzian gradient: All function $f_i(\cdot)$'s are with L-Lipschitzian gradients.*

$$\|\nabla f_i(y) - \nabla f_i(x)\| \leq L \|y - x\|, \quad \forall x, y \in R^N, i \in \{1, \ldots, n\}.$$

- *Bounded gradient variance:*

$$\mathbb{E}_{\xi \sim D_i} \|\nabla F_i(x; \xi) - \nabla f_i(x)\|^2 \leq \sigma^2, \quad \forall i, \forall x \in R^N.$$

- *Bounded gradient divergence:*

$$\frac{1}{n} \sum_{i=1}^n \mathbb{E} \left\| \nabla f_i(x) - \overline{\nabla f}(x) \right\|^2 \leq \epsilon^2, \quad \forall i, \forall x \in R^N.$$

*where $\overline{\nabla f}(x) = \frac{1}{n} \nabla f_i(x^i)$.*

- *Unbiased stochastic compression: The stochastic compression operator $C(\cdot)$ is unbiased, which satisfies $\mathbb{E}\left[C(b)\right] = b$ for any $b \in \mathbb{R}^d$.[1]*

- *Bounded compression error: $\left\|b_t^i - C(b_t^i)\right\|^2 \le \beta \left\|b_t^i\right\|^2$ for $0 \le \beta \le 1$.*

In Assumption 1, the bounded variance $\sigma^2$ captures the stochastic gradient noise, and the bounded gradient divergence $\epsilon^2$ characterizes the degree of heterogeneity of data distribution across all nodes. The value of $\epsilon^2$ equals to zero if all nodes share the same data distribution. The parameter $\beta$ in the bounded compression error is important to capture the degree of compression. If an algorithm allows a larger $\beta$, it means that this algorithm can handle a higher degree of compression.

First, we provide the general convergence upper bound for DEFD-PSGD Algorithm 1 as follows.

**Theorem 1.** *When assumption 1 holds, and $a$, $b$ and $c$ are some positive constants, let $\beta < \frac{(1-\rho)^2}{\mu^2(1+a)(1+b)+\mu^2\gamma^2(1+a)(1+b^{-1})(1+c)+\gamma^2(1+a)(1-\rho)^2}$, $\gamma \ge \frac{b}{2}$ and $\eta$ satisfies $1 - B_1 \ge 0$, Algorithm 1 ensures that*

$$\frac{1}{T}\sum_{t=0}^{T-1}\left(\mathbb{E}\left[\nabla f\left(\frac{X_t \mathbf{1}_n}{n}\right)\right] + (1 - B_1)\mathbb{E}\left[\overline{\nabla f}\left(X_t\right)\right]\right)$$

$$\le \frac{2(f(0) - f^*)}{\eta T} + \left(\frac{\eta L}{n} + \frac{8\eta^2 L^2}{(1-\rho)^2} + 8\eta C_1(1+\gamma^2)\left(\frac{\eta L^2}{1-\rho^2} + \frac{L}{2n}\right)\right)\sigma^2$$

$$+ \left(\frac{8\eta^2 L^2}{(1-\rho)^2} + 8\eta C_1(1+\gamma^2)\left(\frac{\eta L^2}{1-\rho^2} + \frac{L}{2n}\right)\right)\epsilon^2, \tag{2}$$

*where $\mu = \max\limits_{i=2,3,\ldots,n}|\lambda_i - 1|$, $f(0)$ is the initial model parameters which is the same among all the nodes and $f^*$ is the true minimum of function $f$, and*

$$C_1 = \frac{\beta(\mu^2(1+a)(1+b^{-1})(1+c) + 2(1+a^{-1})(1-\rho)^2)}{(1-\rho)^2 - \beta\mu^2(1+a)(1+b) - \beta\gamma^2(\mu^2(1+a)(1+b^{-1})(1+c^{-1}) + 2(1+a^{-1})(1-\rho)^2)}, \tag{3}$$

*and*

$$B_1 = \left(\eta L + \frac{4\eta^2 L^2}{(1-\rho)^2} + 4\eta C_1(1+\gamma^2)\left(\frac{\eta L^2}{1-\rho^2} + \frac{L}{2n}\right)\right). \tag{4}$$

The major challenge in the proof of Theorem 1 is to upper bound the residual error term, $\sum_{t=0}^{T-1}\sum_{i=1}^{n}\mathbb{E}_t\left[\left\|e_{t+1}^i\right\|^2\right]$. We develop a novel technique to obtain the following iterative relation,

$$\sum_{t=0}^{T-1}\sum_{i=1}^{n}\mathbb{E}_t\left[\left\|e_{t+1}^i\right\|^2\right] \le A_1\sum_{t=0}^{T-1}\sum_{i=1}^{n}\mathbb{E}_t\left[\left\|e_t^i\right\|^2\right] + A_2\sum_{t=0}^{T-1}\sum_{i=1}^{n}\mathbb{E}_t\left[\left\|G_i(X_t^i)\right\|^2\right], \tag{5}$$

where $A_1$ and $A_2$ are some positive constants related to the system parameters and given in (B.23) and (B.24) in Appendix B. In addition, unlike the strict assumptions used by Koloskova et al. (2019) such as bounded gradients and consistent gradient averages across iterations, the proof of Theorem 1 only uses Assumption 1, which is widely used in distributed optimization and federated learning literature.

From Theorem 1, we observe that the only term influenced by $\beta$ is the value of $C_1$. The parameter $\beta$ is defined such that $\left\|b_t^i - C(b_t^i)\right\|^2 \le \beta\left\|b_t^i\right\|^2$, where $0 \le \beta \le 1$. It can be seen that the convergence upper bound increases as $\beta$ grows, indicating that higher compression errors lead to a higher convergence bound. However, a larger upper limit for $\beta$ also implies that the algorithm can handle higher levels of compression. This will be discussed in detail later. In order to get a better understanding of the impact of different parameters in DEFD-PSGD, by choosing specific values of $a, b, c$, we obtain the following corollary.

---

[1]We emphasize that the unbiased compressor assumption is only made to be consistent with the assumption in (Tang et al., 2018). In Section 5, we show empirically that DEFD-PSGD can outperform state-of-the-art algorithms when biased compression such as top-$k$ is used.

**Corollary 1.** *When Assumption 1 holds, and* $a = \frac{\sqrt{2}\gamma(1-\rho)}{\mu(\sqrt{2}\gamma+1)}$, $b = \sqrt{2}\gamma$ *and* $c = 1$, *let* $\gamma > 0$, $\beta < \frac{(1-\rho)^2}{(\mu(\sqrt{2}\gamma+1)+\sqrt{2}\gamma(1-\rho))^2}$ *and* $\eta$ *satisfies* $1 - B_2 \geq 0$, *Algorithm 1 ensures that*

$$
\frac{1}{T}\sum_{t=0}^{T-1}\left(\mathbb{E}\left[\nabla f\left(\frac{X_t\mathbf{1}_n}{n}\right)\right] + (1-B_2)\,\mathbb{E}\left[\overline{\nabla f}\left(X_t\right)\right]\right)
$$

$$
\leq \frac{2(f(0)-f^*)}{\eta T} + \left(\frac{\eta L}{n} + \frac{8\eta^2 L^2}{(1-\rho)^2} + \frac{8\sqrt{2}\eta(1+\gamma^2)}{\gamma}C_2\left(\frac{\eta L^2}{1-\rho^2} + \frac{L}{2n}\right)\right)\sigma^2
$$

$$
+ \left(\frac{8\eta^2 L^2}{(1-\rho)^2} + \frac{8\sqrt{2}\eta(1+\gamma^2)}{\gamma}C_2\left(\frac{\eta L^2}{1-\rho^2} + \frac{L}{2n}\right)\right)\epsilon^2, \tag{6}
$$

*where* $\mu = \max\limits_{i=2,3,\ldots,n}|\lambda_i - 1|$, $f(0)$ *is the initial model parameters which is the same among all the nodes and* $f^*$ *is the true minimum of function* $f$, *and*

$$
C_2 = \frac{\beta(\mu+(1-\rho))(\mu(\sqrt{2}\gamma+1)+\sqrt{2}\gamma(1-\rho))}{(1-\rho)^2 - \beta(\mu(\sqrt{2}\gamma+1)+\sqrt{2}\gamma(1-\rho))^2}, \tag{7}
$$

*and*

$$
B_2 = \left(\eta L + \frac{4\eta^2 L^2}{(1-\rho)^2} + \frac{4\sqrt{2}\eta(1+\gamma^2)}{\gamma}C_2\left(\frac{\eta L^2}{1-\rho^2} + \frac{L}{2n}\right)\right). \tag{8}
$$

**The upper bound of** $\beta$. From Corollary 1, it can be seen that with a proper choice of $a, b$ and $c$, then we have

$$
\beta < \frac{(1-\rho)^2}{(\mu(\sqrt{2}\gamma+1)+\sqrt{2}\gamma(1-\rho))^2}, \tag{9}
$$

where $\gamma > 0$. It can be seen that for a given decentralized network topology, we can always find a unique value $\gamma_0 = \frac{\mu}{\sqrt{2}(\mu+(1-\rho))}$ so that when $\gamma \in (0, \gamma_0]$, the upper bound of $\beta$ in (9) is larger than the one given in the DCD-PSGD algorithm by Tang et al. (2018), which is $\beta < \frac{(1-\rho)^2}{4\mu^2}$. This means that DEFD-PSGD allows more compression compared to DCD-PSGD under the same network topology. Numerically, this can also be seen from Table 1, where the random-$y$ network means that we choose $y$ connected nodes uniformly at random for each node and guarantee that the connectivity matrix $W$ satisfies Assumption 1.

Table 1: The upper bound of $\beta$ with different network topologies. DCD-PSGD has a consistent bound for each network topology. DEFD-PSGD has a upper bound range with different choice of $\gamma \in (0, 1]$ for each network topology.

| Network Topology | DCD-PSGD | DEFD-PSGD |
|---|---|---|
| Ring | $8.49e^{-32}$ | $(5.83e^{-32}, 3.40e^{-31}]$ |
| Random-4 | 0.04 | $(0.0024, 0.0162]$ |
| Random-9 | 0.051 | $(0.22, 0.205]$ |
| Fully Connected | 0.25 | $(0.068, 0.99]$ |

**Comparison between DEFD-PSGD and DCD-PSGD.** In Theorem 1, if we choose $\gamma = 0$ and $a = b = c = 1$ in (2), we can recover the convergence upper bound for DCD-PSGD up to some constants. Comparing this DCD-PSGD convergence upper bound with (6) in Corollary 1, if we choose the optimal $\gamma \in (0, \gamma_0]$ to minimize (6), we find that (6) is smaller than that of DCD-PSGD with a given real connectivity matrix $W$ and $\beta$. In addition, the minima of (6) can also be found in the range of $\gamma \in (0, \gamma_0]$. One example can be found in Figure A.2a in Appendix A.2. In addition, under the same setting, in Figure A.2b, we plot the upper bound of DCD-PSGD and (6) for different value of $\beta$, where (6) is optimized over $\gamma$. Here, it can be seen that 1) DCD-PSGD goes to infinity when $\beta = 0.004$ while the proposed DEFD-PSGD does not and 2) we show empirically that (6) is smaller than the upper bound of DCD-PSGD.

**Comparison between DEFD-PSGD and CHOCO-PSGD.** As discussed in Section 1, CHOCO-PSGD and related algorithms propose another approach using residual error feedback. Even though these algorithms can tolerate a higher degree of gradient compression, they cannot handle a high degree of data heterogeneity. In order to illustrate this, we rewrite the CHOCO-PSGD algorithm in terms of error feedback to explicitly show the difference compared to DEFD-PSGD. From Algorithm A.1 in Appendix A.4, it can be seen that the models are not synchronized among neighboring nodes. This means that each node computes a "compressed" copy of the model for each connected node, e.g., node $i$ compute $\hat{x}_{t+1}^{i,j}$, which is the computed model of its neighboring node $j$ at node $i$ where $\hat{x}_{t+1}^{i,j}$ is not the same as $x_{t+1}^{j}$. Hence, in order to make CHOCO-PSGD converge, it needs to introduce a parameter $\gamma'$ to control the model update in each iteration, in which $\gamma'$ is much less than 1 in practice. This reduces the impact of model aggregation among connected nodes significantly so that when data is highly heterogeneous, the CHOCO-PSGD may diverge. The role of $\gamma$ in the proposed DEFD-PSGD is similar to $\gamma'$ in CHOCO-PSGD, in that $\gamma$ can also control the impact of model aggregation among neighboring nodes in DEFD-PSGD. However, due to the fact that the models are perfectly synchronized among connected nodes, $\gamma$ can be much larger compared to $\gamma'$ in CHOCO-PSGD. This is shown in Table A.1 in Appendix A.2.

**Corollary 2.** *Let learning rate $\eta = \left(L + \frac{\sigma\sqrt{T}}{\sqrt{n}} + \epsilon^{\frac{2}{3}}T^{\frac{1}{3}}\right)^{-1}$ in Algorithm 1, according to Theorem 1, if we have $\beta < \frac{(1-\rho)^2}{\mu^2(1+a)(1+b)+\mu^2\gamma^2(1+a)(1+b^{-1})(1+c)+\gamma^2(1+a)(1-\rho)^2}$ and treat $\gamma$ as constant, the convergence rate becomes*

$$\frac{1}{T}\sum_{t=0}^{T-1}\mathbb{E}\left[\nabla f\left(\frac{X_t\mathbf{1}_n}{n}\right)\right] = O\left(\frac{\sigma}{\sqrt{nT}} + \frac{\epsilon^{\frac{2}{3}}}{T^{\frac{2}{3}}} + \frac{1}{T}\right), \tag{10}$$

The Corollary 2 shows that we can achieve the same convergence rate as Tang et al. (2018); Koloskova et al. (2019). Moreover, the dominant term of convergence rate is $O\left(\frac{1}{\sqrt{nT}}\right)$, which is consistent with the convergence rate of centralized SGD. In addition, the first two dominant terms are $O\left(\frac{\sigma}{\sqrt{nT}} + \frac{\epsilon^{\frac{2}{3}}}{T^{\frac{2}{3}}}\right)$, which are consistent with D-PSGD in Lian et al. (2017).

## 5 EXPERIMENTS

In this section, we provide the experimental setup including details on the models, datasets, and compression schemes used. We include two baseline algorithms, DCD-PSGD (Tang et al., 2018) and CHOCO-PSGD (Koloskova et al., 2019), for comparison, since they are most representative. Comparisons between DEFD-PSGD and other algorithms can be found in Appendix A.5.

### 5.1 EXPERIMENTAL SETUP

**Dataset and Models.** In our experiments, we evaluate the proposed DEFD-PSGD alongside two baseline algorithms, DCD-PSGD and CHOCO-PSGD, using two popular datasets, FashionMNIST (Xiao et al., 2017) and CIFAR10 (Krizhevsky & Hinton, 2009). For the FashionMNIST dataset, we use a two-layer neural network. For CIFAR10 dataset, we employ a Convolutional Neural Network (CNN) consisting of two convolutional layers, each paired with a max-pool layer (with a $3 \times 3$ kernel padding, 32 filters and a $2 \times 2$ max-pool), followed by three fully connected layers (with sizes 256, 64, 10) (Wang et al., 2023). We apply ReLU activation functions to all layers except the final output layer and use Kaiming initialization (He et al., 2015) for the initial model parameters. To ensure a fair comparison, we provide the average results from four runs with different random seeds.

**Data distribution.** To control the degree of heterogeneity for the data distributions across the nodes, we apply the Dirichlet distribution which is parameterized by $\alpha$. The details about Dirichlet distribution is described in (Tzu-Ming Harry et al., 2019). Specifically, if the Dirichlet parameter $\alpha$ approaches zero, the data distribution becomes extremely heterogeneous, which means the data distribution on each node mainly contains one class of data. Otherwise, if the Dirichlet parameter $\alpha$ goes to infinity, the local data distribution tends to be the same across all nodes. In our experiments, we employ the Dirichlet parameter $\alpha = 0.05$ to evaluate the performance of different algorithms

on highly heterogeneous data. We provide further experimental results with additional Dirichlet parameters' values in Appendix A.5.

**Network topology.** We consider a decentralized network topology where each node communicates only with their neighboring nodes. The real connectivity matrix $W$ is generated based on the total number of nodes and the number of connected nodes per node. We select the neighboring nodes for each node uniformly at random and ensure that the real connectivity matrix $W$ is a symmetric doubly stochastic matrix, satisfying Assumption 1. More specifically, in our experiments, we use a $20 \times 20$ symmetric doubly stochastic matrix, where each node is connected to $4$ other nodes.

**Compression.** In our experiments, we employ both unbiased and biased compression techniques to evaluate the performance of the proposed DEFD-PSGD and other algorithms. For the unbiased compression, we use element-wise random quantization, as described in Zhang et al. (2017). Specifically, each element is randomly quantized into one of the two closest quantization levels. The probability for each level is calculated based on the distance between the element and the quantization levels normalized by the distance between the two quantization levels. For biased compression, we apply the Top-$k$ compression (Stich et al., 2018; Alistarh et al., 2018), which selects the top $k$ fraction of all elements according to the magnitude and sets other elements to zero. For quantization, we apply $4$-bit and $6$-bit random quantization for FashionMNIST dataset and $6$-bit and $8$-bit random quantization for CIFAR10 dataset. For top-$k$ compression, we use top-$k$ ratio of $0.1$ and $0.2$ for both FashionMNIST and CIFAR10 dataset. With these parameters, all the algorithms used in our experiments have been shown to converge empirically.

## 5.2 EXPERIMENTAL RESULTS

In this section, we provide the experimental results for FashionMNIST and CIFAR10 dataset with different compression methods. For each experiment with different compression schemes, to find the proper value of hyper-parameters in order to achieve the best performance of each algorithm, we perform a grid search for every hyper-parameter in each algorithm. The details about finding proper hyper-parameters is shown in Appendix A.1. Here, we would like to emphasize the the choice of $\gamma$ depends on the range of $\beta$ (see Table 1) for the proposed DEFD-PSGD.

Table 2: Test accuracy of both top-$k$ and quantization methods using FashionMNIST dataset. The results are averaged over four experiments with different initial model parameters, we list the averaged accuracy and the standard variance.

| FashionMNIST | Top-$k$ (%) | | Quantization | |
|---|---|---|---|---|
| Algorithm | 10% | 20% | 4 bits | 6 bits |
| DCD-PSGD | $75.43 \pm 1.39$ | $78.18 \pm 1.21$ | $71.05 \pm 0.85$ | $79.59 \pm 1.07$ |
| CHOCO-PSGD | $76.21 \pm 1.73$ | $76.04 \pm 1.71$ | $75.97 \pm 1.29$ | $76.31 \pm 1.58$ |
| DEFD-PSGD (ours) | $\mathbf{77.63} \pm 1.27$ | $\mathbf{79.30} \pm 1.46$ | $\mathbf{78.84} \pm 0.63$ | $\mathbf{80.21} \pm 0.69$ |

Table 3: Test accuracy of both top-$k$ and quantization methods using CIFAR10 dataset. The results are averaged over four experiments with different initial model parameters, we list the averaged accuracy and the standard variance.

| CIFAR10 | Top-$k$ (%) | | Quantization | |
|---|---|---|---|---|
| Algorithm | 10% | 20% | 6 bits | 8 bits |
| DCD-PSGD | $62.14 \pm 0.84$ | $67.05 \pm 0.78$ | $53.48 \pm 2.21$ | $66.57 \pm 1.01$ |
| CHOCO-PSGD | $63.27 \pm 1.42$ | $63.26 \pm 1.78$ | $63.28 \pm 1.21$ | $63.66 \pm 1.20$ |
| DEFD-PSGD (ours) | $\mathbf{67.27} \pm 0.89$ | $\mathbf{69.27} \pm 0.89$ | $\mathbf{69.81} \pm 1.24$ | $\mathbf{70.50} \pm 0.91$ |

From Tables 2, 3, and Figures 2, 3, our general observation is that the proposed DEFD-PSGD provides the highest test accuracy compared to both baselines. In particular, first, we can see that DCD-PSGD does not perform well when the degree of compression is high. For example, for the FashionMNIST dataset and when top-$k$ is used, it can be observed that the test accuracy of DCD-PSGD drops from $78.18$ to $75.43$ when $k$ reduces from $0.2$ to $0.1$. Second, CHOCO-PSGD can

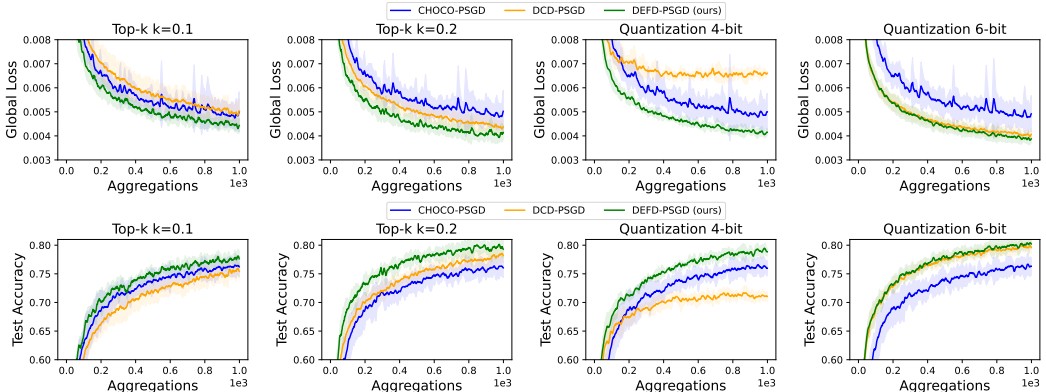

Figure 2: Global loss and test accuracy for FashionMNIST dataset with different compression schemes: Top-$k$ compression with $k = 0.1$ and $k = 0.2$, and random quantization with 4-bit and 6-bit.

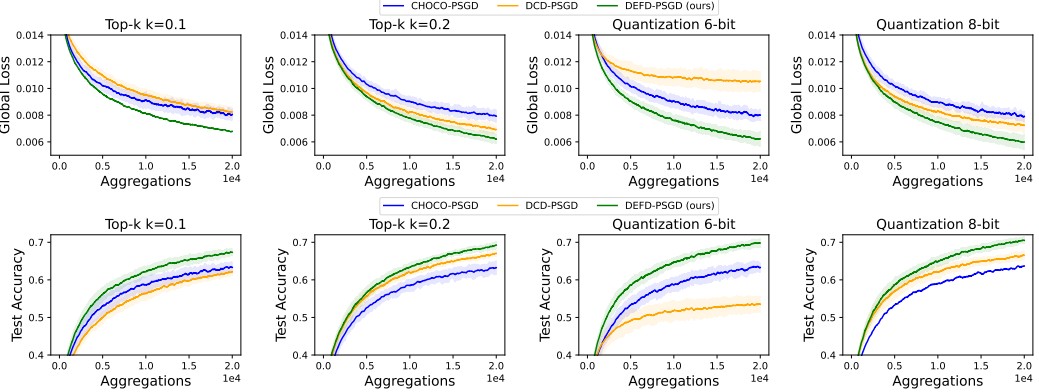

Figure 3: Global loss and test accuracy for CIFAR10 dataset with different compression schemes: Top-$k$ compression with $k = 0.1$ and $k = 0.2$, and random quantization with 6-bit and 8-bit.

indeed handle a high degree of gradient compression. For instance, for the FashionMNIST dataset with top-$k$ compression, the test accuracy stays almost the same as $k$ drops from $0.2$ to $0.1$. Third, as we can see in Figure A.4, CHOCO-PSGD cannot handle a high degree of data heterogeneity when the consensus step is not small enough. Here, when we choose the optimal consensus step, it can be seen that the proposed DEFD-PSGD can still outperform CHOCO-PSGD. This is consistent with our intuition. The reason is that due to a small consensus step parameter $\gamma'$ in CHOCO-PSGD, nodes cannot obtain enough gradient update information from neighbors, while DEFD-PSGD can use the discount parameter $\gamma$ to effectively balance between gradient compression and data heterogeneity.

## 6 CONCLUSION

In this paper, we address the challenging problem of decentralized learning, particularly in scenarios involving high gradient compression and significant data heterogeneity. To tackle these challenges, we propose the DEFD-PSGD algorithm, which introduces controllable error feedback to effectively manage gradient compression and data heterogeneity while maintaining communication efficiency. In addition, we develop novel proof techniques to establish a convergence upper bound under more relaxed assumptions. Finally, our experimental results align with the theoretical analysis and demonstrate that DEFD-PSGD outperforms other state-of-the-art decentralized learning algorithms.

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

# Appendix

# A  ADDITIONAL DETAILS ON EXPERIMENTS AND FURTHER EMPIRICAL STUDIES

## A.1  HYPER-PARAMETERS AND EXAMPLE NETWORK TOPOLOGY

To achieve the best performance for the proposed DEFD-PSGD algorithm and baseline algorithms, we need to properly choose the hyper-parameters such as learning rate $\eta$ and discount coefficient $\gamma$. For all the algorithms in the experiments, we use a grid search process to find the proper learning rate $\eta \in \{10^{-3}, 10^{-2.75}, 10^{-2.5}, \ldots, 10^{-0.25}, 10^{0}\}$. To find the value of discount coefficient $\gamma$ that has the best performance for DEFD-PSGD, we compare the results with different discount coefficient values $\gamma \in \{0.05, 0.1, 0.15, \ldots, 0.95, 1\}$. Before running the experiments, we perform a grid search for all hyper-parameters for each algorithm with a given compression level and Dirichlet parameter. Therefore, we have different values of learning rate $\eta$ and $\gamma$ for different scenarios of gradient compression and data heterogeneity. Also, we provide one example of the network topology, which we are using in our experiments in Figure A.1.

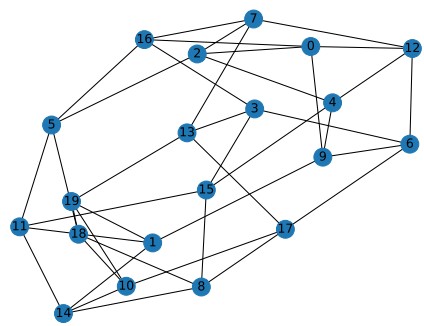

Figure A.1: Network Topology with total nodes of 20 and each connected 4 nodes. Each edge represents a two way communication link.

## A.2  CONVERGENCE UPPER BOUND COMPARISON BETWEEN DEFD-PSGD AND DCD-PSGD

We consider a network with $n = 20$ nodes and each is connected to $4$ nodes. In Theorem 1, if we let $1 - B_1 \geq 0$, $C_1$ is the only part which impacts the value of convergence upper bound. Here, we compare the value of $C_1$ with different values of $\gamma$ or $\beta$ in Figure A.2. In Figure A.2a, we compare the two upper bounds when $\beta = 0.002$. In Figure A.2b, we plot the upper bound of DCD-PSGD and (6) for different value of $\beta$, where (6) is optimized over $\gamma$.

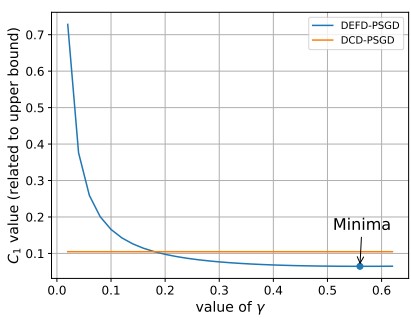

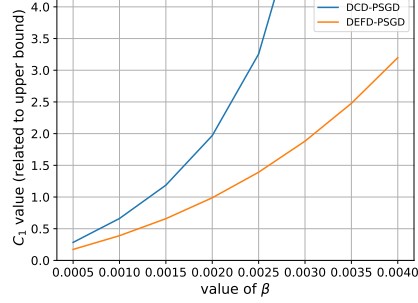

(a) $C_1$ values with $\gamma_0 = 0.62$. The dot represents the minima value. Given $\beta = 0.002$.

(b) $C_1$ values as function of $\beta$.

Figure A.2: Convergence upper bound comparison between DEFD-PSGD and DCD-PSGD.

### A.3 CHOCO-PSGD AS ERROR FEEDBACK

---

**Algorithm A.1:** CHOCO-PSGD as Error Feedback

---

**Input:** $\eta > 0, W, T$

**Output:** $\{x_t^i\}$

**Initialize:** $e_0^i \leftarrow 0, \forall i, \{x_0^i\}_{i=1}^n = x_0, \{\hat{x}_0^i\}_{i=1}^n = 0, \{\hat{x}_0^{i,j}\}_{j=1}^n = 0$

1 **for** $t \leftarrow 0, 1, 2, ..., T-1$ **do**

2    **for** *each node* $i \leftarrow 1, 2, ..., n$ **do**

3       Randomly sample $\xi_t^i$ from local dataset;

4       $x_t^i \leftarrow x_{t-\frac{1}{2}}^i + \gamma' \sum_{j=1}^n w_{ij} \left( \hat{x}_t^{i,j} - \hat{x}_t^i \right)$

5       $b_t^i \leftarrow x_t^i - x_{t-1}^i + e_t^i$;

6       $v_t^i \leftarrow C_t^i(b_t^i)$;

7       $e_{t+1}^i \leftarrow b_t^i - v_t^i$;

8       Send $v_t^i$ and receive $v_t^j$;

9    **for** *each node* $i \leftarrow 1, 2, ..., n$ **do**

10       $\hat{x}_{t+1}^i \leftarrow \hat{x}_t^i + v_t^i$

11       **for** *each* $j, w_{ij} \neq 0$ **do**

12          $\hat{x}_{t+1}^{i,j} \leftarrow \hat{x}_t^{i,j} + v_t^j$

13       Randomly sample $\xi_t^i$ from $D_i$

14       $x_{t+\frac{1}{2}}^i \leftarrow x_t^i - \eta \nabla F_i \left( x_t^i; \xi_t^i \right)$

---

### A.4 STUDY THE IMPACTS OF $\gamma$ IN DEFD-PSGD AND $\gamma'$ IN CHOCO-PSGD

We provide further experimental results to study the impact of $\gamma$ in DEFD-PSGD and $\gamma'$ in CHOCO-PSGD. The network topology is given in Figure A.1 and Dirichlet parameter $\alpha = 0.05$. We choose values for $\gamma$ and $\gamma'$ in the range $(0, 1]$, with a step size of $0.1$. In Figure A.3, the aqua line represents DCD-PSGD with direct error feedback applied.

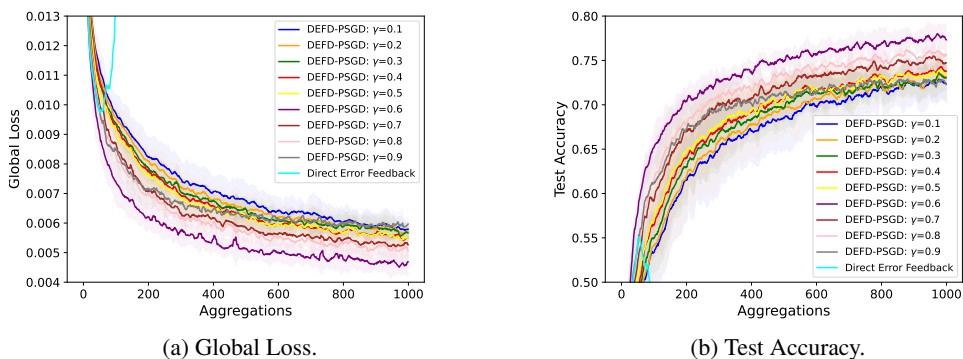

(a) Global Loss.         (b) Test Accuracy.

Figure A.3: Experimental results with different values of $\gamma$ in Algorithm 1. Dirichlet parameter $\alpha = 0.05$.

We also provide Table A.1, to show the impact of different values of $\gamma$ in DEFD-PSGD and $\gamma'$ in CHOCO-PSGD under the same compression method and data distribution. In Table A.1, we compare the test accuracy of CHOCO-PSGD with different value of $\gamma'$ and DEFD-PSGD with different value of $\gamma$. We show that the choice range of $\gamma$ can be larger than the range of $\gamma'$ without causing divergence. Although $\gamma$ and $\gamma'$ serve as different coefficients in DEFD-PSGD and CHOCO-PSGD, we illustrate that DEFD-PSGD offers greater flexibility in selecting the coefficient.

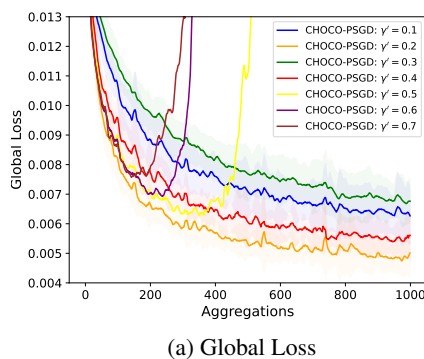
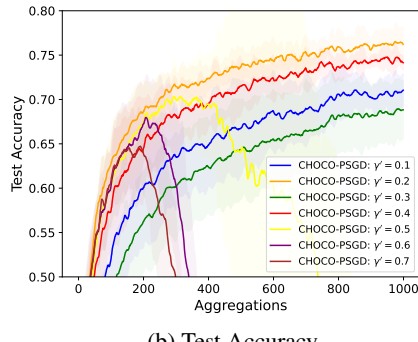

(a) Global Loss

(b) Test Accuracy.

Figure A.4: Experimental results with different value of $\gamma'$ in Algorithm A.1. Dirichlet parameter $\alpha = 0.05$.

Table A.1: The comparison between $\gamma$ in Algorithm 1 and $\gamma'$ in Algorithm A.1, given Dirichlet parameter $\alpha = 0.05$, top-$k$ compression with $k = 0.1$ and learning rate $\eta = 0.056$.

| $\gamma'/\gamma$ | CHOCO-PSGD | DEFD-PSGD | $\gamma'/\gamma$ | CHOCO-PSGD | DEFD-PSGD |
|---|---|---|---|---|---|
| 0.1 | $71.04 \pm 1.86$ | $72.35 \pm 2.10$ | 0.6 | Diverge | $\textbf{77.33} \pm 1.66$ |
| 0.2 | $\textbf{76.22} \pm 1.73$ | $73.00 \pm 1.26$ | 0.7 | Diverge | $74.73 \pm 0.98$ |
| 0.3 | $68.83 \pm 3.31$ | $73.08 \pm 1.55$ | 0.8 | Diverge | $75.67 \pm 1.24$ |
| 0.4 | $74.14 \pm 0.82$ | $73.78 \pm 0.91$ | 0.9 | Diverge | $72.28 \pm 1.83$ |
| 0.5 | Diverge | $73.36 \pm 1.28$ | 1.0 | Diverge | Diverge |

In Figure A.3, we show that directly adding error feedback to DCD-PSGD does not lead to convergence when the data distribution is highly heterogeneous. Moreover, by applying discounted error feedback, we can find an appropriate discount coefficient $\gamma$ that optimizes the performance of DEFD-PSGD. In Figure A.4, we show that the value of $\gamma'$ for CHOCO-PSGD cannot be too large when the degree of data heterogeneity is high.

## A.5 ADDITIONAL EXPERIMENTAL RESULTS

Before presenting additional experimental results, we first discuss the impact of data heterogeneity. In summary, a larger Dirichlet parameter $\alpha$ results in a more homogeneous data distribution. For the case of $\alpha = 0.01$, each node predominantly contains a single class of data, with only a few samples from the other classes. In contrast, for $\alpha = 0.5$, each node has data from nearly all classes, with one or two classes being the majority.

Table A.2: The comparison between DEFD-PSGD, DCD-PSGD and CHOCO-PSGD when the Dirichlet parameter $\alpha = 0.01$ with random quantization and top-$k$ compression.

| CIFAR10 | Top-$k$ (%) | | Quantization | |
|---|---|---|---|---|
| Algorithm | 10% | 20% | 4 bits | 6 bits |
| DCD-PSGD | $64.82 \pm 5.01$ | $69.49 \pm 5.39$ | $65.89 \pm 2.98$ | $71.26 \pm 6.10$ |
| CHOCO-PSGD | $64.02 \pm 7.00$ | $63.63 \pm 7.06$ | $64.26 \pm 7.17$ | $63.99 \pm 6.69$ |
| DEFD-PSGD (ours) | $69.21 \pm 5.78$ | $70.61 \pm 6.06$ | $72.14 \pm 7.17$ | $72.75 \pm 6.32$ |

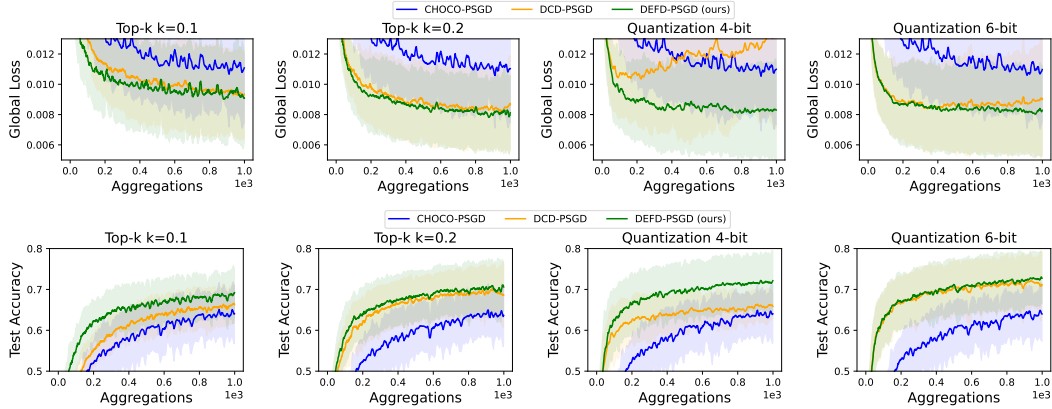

Figure A.5: Global loss and test accuracy for FashionMNIST dataset with different compression schemes: top-$k$ compression with $k = 0.1$ and $k = 0.2$, and random quantization with 4-bit and 6-bit. Dirichlet parameter $\alpha = 0.01$.

Table A.3: The comparison between DEFD-PSGD, DCD-PSGD and CHOCO-PSGD when the Dirichlet parameter $\alpha = 0.5$ with random quantization and Top-$k$ compression scheme.

| CIFAR10 | Top-$k$ (%) | | Quantization | |
|---|---|---|---|---|
| Algorithm | 10% | 20% | 4 bits | 6 bits |
| DCD-PSGD | $80.80 \pm 0.72$ | $82.29 \pm 0.47$ | $77.51 \pm 0.87$ | $83.15 \pm 0.67$ |
| CHOCO-PSGD | $83.11 \pm 0.68$ | $83.46 \pm 0.59$ | $83.41 \pm 0.56$ | $83.45 \pm 0.46$ |
| DEFD-PSGD (ours) | $82.59 \pm 0.70$ | $83.55 \pm 0.59$ | $82.51 \pm 0.93$ | $83.65 \pm 0.52$ |

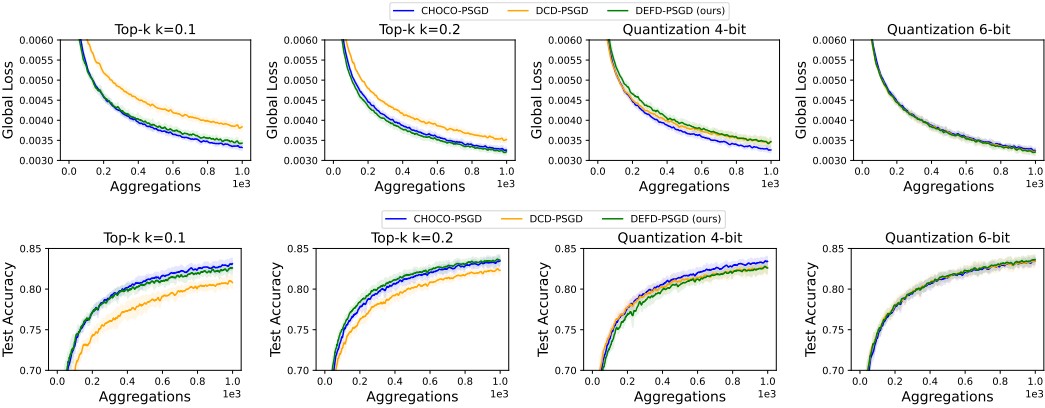

Figure A.6: Global loss and test accuracy for FashionMNIST dataset with different compression schemes: top-$k$ compression with $k = 0.1$ and $k = 0.2$, and random quantization with 4-bit and 6-bit. Dirichlet parameter $\alpha = 0.5$.

In Figure A.5, we present the comparison results of DEFD-PSGD, DCD-PSGD and CHOCO-PSGD for Dirichlet parameter $\alpha = 0.01$, which represents an extremely high level of data heterogeneity. We demonstrate that DEFD-PSGD can tolerate higher level of compression while maintaining promising performance. In Figure A.6, we compare the results for Dirichlet parameter $\alpha = 0.5$, where the data distribution becomes more homogeneous. We show that DEFD-PSGD and CHOCO-PSGD offer similar performance levels for biased compression. Moreover, all three algorithms achieve a similar performance level with unbiased compression.

We also provide a comparison with other state-of-art algorithms, such as AdaG-PSGD and Comp Q-SADDLe. In Figure A.7, we set Dirichlet parameter $\alpha = 0.05$ and perform a grid search for consensus value of AdaG-PSGD and Comp Q-SADDLe within the range of $(0, 1]$. or other coefficients mentioned in AdaG-PSGD and Comp Q-SADDLe algorithms, we use the values suggested in (Aketi et al., 2024; Choudhary et al., 2024). We observe that the performance of AdaG-PSGD and Comp Q-SADDLe could be varied on the choice of coefficient values.

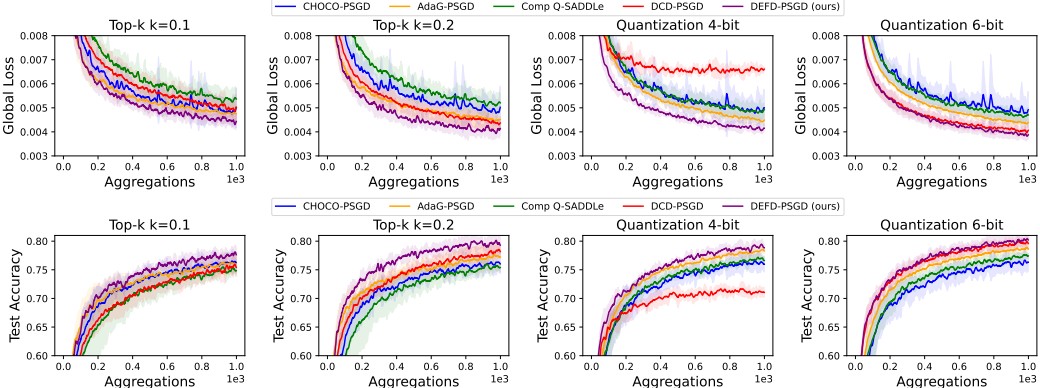

Figure A.7: Global loss and test accuracy for FashionMNIST dataset with different compression schemes: top-$k$ compression with $k = 0.1$ and $k = 0.2$, and random quantization with $4$-bit and $6$-bit.

In Figure A.7, we present a comparison of experimental results for CHOCO-PSGD, AdaG-PSGD, Comp Q-SADDLe, DCD-PSGD and DEFD-PSGD. We set Dirichlet parameter $\alpha = 0.05$ and apply the network topology in Figure A.1. We demonstrate that the proposed DEFD-PSGD outperforms other algorithms in terms of test accuracy and global loss with biased compression. For unbiased compression, while DEFD-PSGD still has the best level of performance, AdaG-PSGD offers promising performance under high level of compression and DCD-PSGD achieves similar level of performance compared to DEFD-PSGD when the compression level is low.

## B    PROOF OF THEOREM 1

We introduce the following notations.

- $X_t = [x_t^1, x_t^2, ..., x_t^n]$.
- $R_t = [r_t^1, r_t^2, ..., r_t^n]$, where $r_t^i = C(b_t^i) - b_t^i$.
- $G(X_t) = [\nabla F_1(x_t^1; \xi_t^1), \nabla F_2(x_t^2; \xi_t^2), ..., \nabla F_n(x_t^n; \xi_t^n)]$.
- Assumption: $\|b_t^i - C(b_t^i)\|^2 \leq \beta \|b_t^i\|^2$ for $0 \leq \beta \leq 1$.

First, following the steps in Algorithm 1, we have the update rule of $x_{t+1}^i$ as follows,

$$
\begin{aligned}
x_{t+1}^i &= x_t^i + v_t^i \\
&= x_t^i + b_t^i - e_{t+1}^i \\
&= x_t^i + x_{t+\frac{1}{2}}^i - x_t^i + \gamma e_t^i - e_{t+1}^i \\
&= x_{t+\frac{1}{2}}^i + \gamma e_t^i - e_{t+1}^i \\
&= \sum_{j=1}^n w_{ij} x_t^j - \eta G_i(x_t^i) + \gamma e_t^i - e_{t+1}^i \\
&= \sum_{j=1}^n w_{ij} x_t^j - \eta G_i(x_t^i) + \gamma e_t^i - (b_t^i - C(b_t^i)) \\
&= \sum_{j=1}^n w_{ij} x_t^j - \eta G_i(x_t^i) + \gamma e_t^i + C(b_t^i) - b_t^i. \quad\quad\text{(B.1)}
\end{aligned}
$$

We define $M_t = \gamma[e_t^1, \ldots, e_t^n]$. Then, the update rule becomes $X_{t+1} = X_t W - \eta G(X_t) + M_t + R_t$. Then, since $W$ is symmetric and doubly stochastic matrix, we have $W \mathbf{1}_n = \mathbf{1}_n$ and

$$
\frac{X_{t+1}\mathbf{1}_n}{n} = \frac{X_t\mathbf{1}_n}{n} - \eta\frac{G(X_t)\mathbf{1}_n}{n} + \frac{M_t\mathbf{1}_n}{n} + \frac{R_t\mathbf{1}_n}{n}. \quad\quad\text{(B.2)}
$$

Now, we can start the derivation by applying the property of Lipschitz continuous gradient.

$$
\mathbb{E}_t\left[f\left(\frac{X_{t+1}\mathbf{1}_n}{n}\right)\right] \leq \mathbb{E}_t\left[f\left(\frac{X_t\mathbf{1}_n}{n}\right)\right] \quad\quad\text{(B.3)}
$$

$$
+ \mathbb{E}_t\left[\left\langle \nabla f\left(\frac{X_t\mathbf{1}_n}{n}\right), -\eta\frac{G(X_t)\mathbf{1}_n}{n} + \frac{M_t\mathbf{1}_n}{n} + \frac{R_t\mathbf{1}_n}{n}\right\rangle\right] \quad\quad\text{(B.4)}
$$

$$
+ \frac{L}{2}\mathbb{E}_t\left[\left\|-\eta\frac{G(X_t)\mathbf{1}_n}{n} + \frac{M_t\mathbf{1}_n}{n} + \frac{R_t\mathbf{1}_n}{n}\right\|^2\right]. \quad\quad\text{(B.5)}
$$

Then, we compute the last two terms, (B.4) and (B.5), separately.

First, we compute (B.4) as follows.

$$
\begin{aligned}
&\mathbb{E}_t\left[\left\langle \nabla f\left(\frac{X_t\mathbf{1}_n}{n}\right), -\eta\frac{G(X_t)\mathbf{1}_n}{n} + \frac{M_t\mathbf{1}_n}{n} + \frac{R_t\mathbf{1}_n}{n}\right\rangle\right] \\
&= \mathbb{E}_t\left[\left\langle \nabla f\left(\frac{X_t\mathbf{1}_n}{n}\right), -\eta\frac{G(X_t)\mathbf{1}_n}{n}\right\rangle\right] \\
&\quad + \mathbb{E}_t\left[\left\langle \nabla f\left(\frac{X_t\mathbf{1}_n}{n}\right), \frac{R_t\mathbf{1}_n}{n}\right\rangle\right] + \mathbb{E}_t\left[\left\langle \nabla f\left(\frac{X_t\mathbf{1}_n}{n}\right), \frac{M_t\mathbf{1}_n}{n}\right\rangle\right]
\end{aligned}
$$

$$\overset{(a)}{=} \mathbb{E}_t \left[ \left\langle \nabla f \left( \frac{X_t \mathbf{1}_n}{n} \right), -\eta \mathbb{E}_t \left[ \frac{G(X_t) \mathbf{1}_n}{n} \right] \right\rangle \right]$$

$$= \mathbb{E}_t \left[ \left\langle \nabla f \left( \frac{X_t \mathbf{1}_n}{n} \right), -\eta \overline{\nabla f} \left( X_t \right) \right\rangle \right]$$

$$= -\frac{\eta}{2} \mathbb{E}_t \left[ \left\| \nabla f \left( \frac{X_t \mathbf{1}_n}{n} \right) \right\|^2 \right] - \frac{\eta}{2} \mathbb{E}_t \left[ \left\| \overline{\nabla f} \left( X_t \right) \right\|^2 \right] + \frac{\eta}{2} \mathbb{E}_t \left[ \left\| \nabla f \left( \frac{X_t \mathbf{1}_n}{n} \right) - \overline{\nabla f} \left( X_t \right) \right\|^2 \right]$$

$$= -\frac{\eta}{2} \mathbb{E}_t \left[ \left\| \nabla f \left( \frac{X_t \mathbf{1}_n}{n} \right) \right\|^2 \right] - \frac{\eta}{2} \mathbb{E}_t \left[ \left\| \overline{\nabla f} \left( X_t \right) \right\|^2 \right]$$

$$+ \frac{\eta}{2n^2} \mathbb{E}_t \left[ \left\| \sum_{i=1}^n \nabla f_i \left( \frac{X_t \mathbf{1}_n}{n} \right) - \nabla f_i \left( x_t^i \right)^2 \right\| \right]$$

$$\overset{(b)}{\leq} -\frac{\eta}{2} \mathbb{E}_t \left[ \left\| \nabla f \left( \frac{X_t \mathbf{1}_n}{n} \right) \right\|^2 \right] - \frac{\eta}{2} \mathbb{E}_t \left[ \left\| \overline{\nabla f} \left( X_t \right) \right\|^2 \right]$$

$$+ \frac{\eta}{2n} \sum_{i=1}^n \mathbb{E}_t \left[ \left\| \nabla f_i \left( \frac{X_t \mathbf{1}_n}{n} \right) - \nabla f_i \left( x_t^i \right) \right\|^2 \right]$$

$$\overset{(c)}{\leq} -\frac{\eta}{2} \mathbb{E}_t \left[ \left\| \nabla f \left( \frac{X_t \mathbf{1}_n}{n} \right) \right\|^2 \right] - \frac{\eta}{2} \mathbb{E}_t \left[ \left\| \overline{\nabla f} \left( X_t \right) \right\|^2 \right] + \frac{\eta L^2}{2n} \sum_{i=1}^n \mathbb{E}_t \left[ \left\| \frac{X_t \mathbf{1}_n}{n} - x_t^i \right\|^2 \right]. \quad \text{(B.6)}$$

In (a), we apply

$$\mathbb{E}_t \left[ \frac{R_t \mathbf{1}_n}{n} \right] = \frac{(\mathbb{E}_t \left[ C(B_t) \right] - \mathbb{E}_t \left[ B_t \right]) \mathbf{1}_n}{n} = 0,$$

and

$$\mathbb{E}_t \left[ \frac{M_t \mathbf{1}_n}{n} \right] = \frac{\gamma (\mathbb{E}_t \left[ B_{t-1} \right] - \mathbb{E}_t \left[ C(B_{t-1}) \right]) \mathbf{1}_n}{n} = 0.$$

We use the Jensen's Inequality in (b) of (B.6). The Jensen's inequality can be written as

$$\mathbb{E}_t \left[ \left\| \sum_{i=1}^n \nabla f_i \left( \frac{X_t \mathbf{1}_n}{n} \right) - \nabla f_i \left( x_t^i \right) \right\|^2 \right] \leq n \sum_{i=1}^n \mathbb{E}_t \left[ \left\| \nabla f_i \left( \frac{X_t \mathbf{1}_n}{n} \right) - \nabla f_i \left( x_t^i \right) \right\|^2 \right].$$

Moreover, (c) in (B.6) applies the $L$-Lipschitz continuous gradients property.

Second, we derive the second part (B.5) as follows.

$$\frac{L}{2} \mathbb{E}_t \left[ \left\| -\eta \frac{G(X_t) \mathbf{1}_n}{n} + \frac{M_t \mathbf{1}_n}{n} + \frac{R_t \mathbf{1}_n}{n} \right\|^2 \right]$$

$$= \frac{L}{2} \mathbb{E}_t \left[ \left\| \frac{-\eta G(X_t) \mathbf{1}_n}{n} + \frac{M_t \mathbf{1}_n}{n} \right\|^2 \right] + \frac{L}{2} \mathbb{E}_t \left[ \left\| \frac{R_t \mathbf{1}_n}{n} \right\|^2 \right]$$

$$+ L \mathbb{E}_t \left[ \left\langle -\eta \frac{G(X_t) \mathbf{1}_n}{n} + \frac{M_t \mathbf{1}_n}{n}, \mathbb{E}_t \left[ \frac{R_t \mathbf{1}_n}{n} \right] \right\rangle \right]$$

$$= \frac{\eta^2 L}{2} \mathbb{E}_t \left[ \left\| \frac{G(X_t) \mathbf{1}_n}{n} \right\|^2 \right] + \frac{L}{2} \mathbb{E}_t \left[ \left\| \frac{M_t \mathbf{1}_n}{n} \right\|^2 \right] + \mathbb{E}_t \left[ \left\langle -\eta \frac{G(X_t) \mathbf{1}_n}{n}, \mathbb{E}_t \left[ \frac{M_t \mathbf{1}_n}{n} \right] \right\rangle \right]$$

$$+ \frac{L}{2} \mathbb{E}_t \left[ \left\| \frac{R_t \mathbf{1}_n}{n} \right\|^2 \right]$$

$$= \frac{\eta^2 L}{2} \mathbb{E}_t \left[ \left\| \frac{1}{n} \sum_{i=1}^n \nabla F_i \left( x_t^i; \xi_t^i \right) \right\|^2 \right] + \frac{\gamma^2 L}{2} \mathbb{E}_t \left[ \left\| \frac{1}{n} \sum_{i=1}^n e_t^i \right\|^2 \right] + \frac{L}{2} \mathbb{E}_t \left[ \left\| \frac{1}{n} \sum_{i=1}^n r_t^i \right\|^2 \right]$$

$$= \frac{\eta^2 L}{2} \mathbb{E}_t \left[ \left\| \frac{1}{n} \sum_{i=1}^n \nabla F_i \left( x_t^i; \xi_t^i \right) \right\|^2 \right] + \frac{\gamma^2 L}{2n^2} \sum_{i=1}^n \mathbb{E}_t \left[ \left\| e_t^i \right\|^2 \right] + \frac{L}{2n^2} \sum_{i=1}^n \mathbb{E}_t \left[ \left\| r_t^i \right\|^2 \right]$$

$$+ \frac{\gamma^2 L}{n^2} \sum_{i \neq i'} \mathbb{E}_t \left[ \left\langle \mathbb{E}_t \left[ e_t^i \right], \mathbb{E}_t \left[ e_t^{i'} \right] \right\rangle \right] + \frac{L}{n^2} \sum_{i \neq i'} \mathbb{E}_t \left[ \left\langle \mathbb{E}_t \left[ r_t^i \right], \mathbb{E}_t \left[ r_t^{i'} \right] \right\rangle \right]$$

$$= \frac{\eta^2 L}{2} \mathbb{E}_t \left[ \left\| \frac{1}{n} \sum_{i=1}^n \nabla F_i \left( x_t^i; \xi_t^i \right) \right\|^2 \right] + \frac{\gamma^2 L}{2n^2} \sum_{i=1}^n \mathbb{E}_t \left[ \left\| e_t^i \right\|^2 \right] + \frac{L}{2n^2} \sum_{i=1}^n \mathbb{E}_t \left[ \left\| r_t^i \right\|^2 \right]$$

$$= \frac{\eta^2 L}{2} \mathbb{E}_t \left[ \left\| \frac{1}{n} \sum_{i=1}^n \nabla F_i \left( x_t^i; \xi_t^i \right) - \frac{1}{n} \sum_{i=1}^n \nabla f_i \left( x_t^i \right) + \frac{1}{n} \sum_{i=1}^n \nabla f_i \left( x_t^i \right) \right\|^2 \right]$$

$$+ \frac{\gamma^2 L}{2n^2} \sum_{i=1}^n \mathbb{E}_t \left[ \left\| e_t^i \right\|^2 \right] + \frac{L}{2n^2} \sum_{i=1}^n \mathbb{E}_t \left[ \left\| r_t^i \right\|^2 \right]$$

$$= \frac{\eta^2 L}{2} \mathbb{E}_t \left[ \left\| \frac{1}{n} \sum_{i=1}^n \nabla f_i \left( x_t^i \right) \right\|^2 \right] + \frac{\eta^2 L}{2} \mathbb{E}_t \left[ \left\| \frac{1}{n} \sum_{i=1}^n \left( \nabla F_i \left( x_t^i; \xi_t^i \right) - \nabla f_i \left( x_t^i \right) \right) \right\|^2 \right]$$

$$+ \eta^2 L \mathbb{E}_t \left[ \left\langle \frac{1}{n} \sum_{i=1}^n \left( \mathbb{E}_t \left[ \nabla F_i \left( x_t^i; \xi_t^i \right) \right] - \nabla f_i \left( x_t^i \right) \right), \frac{1}{n} \sum_{i=1}^n \nabla f_i \left( x_t^i \right) \right\rangle \right]$$

$$+ \frac{\gamma^2 L}{2n^2} \sum_{i=1}^n \mathbb{E}_t \left[ \left\| e_t^i \right\|^2 \right] + \frac{L}{2n^2} \sum_{i=1}^n \mathbb{E}_t \left[ \left\| r_t^i \right\|^2 \right]$$

$$= \frac{\eta^2 L}{2} \mathbb{E}_t \left[ \left\| \overline{\nabla f} \left( X_t \right) \right\|^2 \right] + \frac{\eta^2 L}{2} \mathbb{E}_t \left[ \left\| \frac{1}{n} \sum_{i=1}^n \left( \nabla F_i \left( x_t^i; \xi_t^i \right) - \nabla f_i \left( x_t^i \right) \right) \right\|^2 \right]$$

$$+ \frac{\gamma^2 L}{2n^2} \sum_{i=1}^n \mathbb{E}_t \left[ \left\| e_t^i \right\|^2 \right] + \frac{L}{2n^2} \sum_{i=1}^n \mathbb{E}_t \left[ \left\| r_t^i \right\|^2 \right]$$

$$= \frac{\eta^2 L}{2} \mathbb{E}_t \left[ \left\| \overline{\nabla f} \left( X_t \right) \right\|^2 \right] + \frac{\gamma^2 L}{2n^2} \sum_{i=1}^n \mathbb{E}_t \left[ \left\| e_t^i \right\|^2 \right] + \frac{L}{2n^2} \sum_{i=1}^n \mathbb{E}_t \left[ \left\| r_t^i \right\|^2 \right]$$

$$+ \frac{\eta^2 L}{2n^2} \sum_{i=1}^n \mathbb{E}_t \left[ \left\| \nabla F_i \left( x_t^i; \xi_t^i \right) - \nabla f_i \left( x_t^i \right) \right\|^2 \right]$$

$$+ \frac{\eta^2 L}{n^2} \sum_{i=1}^n \sum_{i'=1}^n \sum_{i \neq i'} \left\langle \mathbb{E}_t \left[ \nabla F_i \left( x_t^i; \xi_t^i \right) \right] - \nabla f_i \left( x_t^i \right), \mathbb{E}_t \left[ \nabla F_{i'} \left( x_t^{i'}; \xi_t^{i'} \right) \right] - \nabla f_{i'} \left( x_t^{i'} \right) \right\rangle$$

$$= \frac{\eta^2 L}{2} \mathbb{E}_t \left[ \left\| \overline{\nabla f} \left( X_t \right) \right\|^2 \right] + \frac{\gamma^2 L}{2n^2} \sum_{i=1}^n \mathbb{E}_t \left[ \left\| e_t^i \right\|^2 \right] + \frac{L}{2n^2} \sum_{i=1}^n \mathbb{E}_t \left[ \left\| r_t^i \right\|^2 \right]$$

$$+ \frac{\eta^2 L}{2n^2} \sum_{i=1}^n \mathbb{E}_t \left[ \left\| \nabla F_i \left( x_t^i; \xi_t^i \right) - \nabla f_i \left( x_t^i \right) \right\|^2 \right]$$

$$\leq \frac{\eta^2 L}{2} \mathbb{E}_t \left[ \left\| \overline{\nabla f} \left( X_t \right) \right\|^2 \right] + \frac{\gamma^2 L}{2n^2} \sum_{i=1}^n \mathbb{E}_t \left[ \left\| e_t^i \right\|^2 \right] + \frac{L}{2n^2} \sum_{i=1}^n \mathbb{E}_t \left[ \left\| r_t^i \right\|^2 \right] + \frac{\eta^2 L \sigma^2}{2n}. \tag{B.7}$$

Then, we replace (B.4) and (B.5) with (B.6) and (B.7), and obtain

$$\mathbb{E}_t\left[f\left(\frac{X_{t+1}\mathbf{1}_n}{n}\right)\right] \leq \mathbb{E}_t\left[f\left(\frac{X_t\mathbf{1}_n}{n}\right)\right] - \frac{\eta}{2}\mathbb{E}_t\left[\left\|\nabla f\left(\frac{X_t\mathbf{1}_n}{n}\right)\right\|^2\right] - \frac{\eta}{2}\mathbb{E}_t\left[\left\|\overline{\nabla f}\left(X_t\right)\right\|^2\right]$$

$$+ \frac{\eta L^2}{2n}\sum_{i=1}^n \mathbb{E}_t\left[\left\|\frac{X_t\mathbf{1}_n}{n} - x_t^i\right\|^2\right] + \frac{\eta^2 L}{2}\mathbb{E}_t\left[\left\|\overline{\nabla f}\left(X_t\right)\right\|^2\right]$$

$$+ \frac{\eta^2\sigma^2 L}{2n} + \frac{\gamma^2 L}{2n^2}\sum_{i=1}^n \mathbb{E}_t\left[\left\|e_t^i\right\|^2\right] + \frac{L}{2n^2}\sum_{i=1}^n \mathbb{E}_t\left[\left\|r_t^i\right\|^2\right]. \tag{B.8}$$

Before we continue on the derivation, we rewrite $X_t$ as follows.

$$X_t = X_0 - \eta\sum_{s=0}^{t-1} G(X_s)W^{t-s-1} + \sum_{s=0}^{t-1} M_s W^{t-s-1} + \sum_{s=0}^{t-1} R_s W^{t-s-1}. \tag{B.9}$$

Next, we compute $\sum_{i=1}^n \mathbb{E}_t\left[\left\|\frac{X_t\mathbf{1}_n}{n} - x_t^i\right\|^2\right]$ and obtain

$$\sum_{i=1}^n \mathbb{E}_t\left[\left\|\frac{X_t\mathbf{1}_n}{n} - x_t^i\right\|^2\right]$$

$$\stackrel{(a)}{\leq} 2\sum_{i=1}^n \mathbb{E}_t\left[\left\|\eta\left(\sum_{s=0}^{t-1} G(X_s)W^{t-s-1}e_i - \frac{G(X_s)\mathbf{1}_n}{n}\right)\right\|^2\right]$$

$$+ 2\sum_{i=1}^n \mathbb{E}_t\left[\left\|\left(\sum_{s=0}^{t-1} M_s W^{t-s-1}e_i - \frac{M_s\mathbf{1}_n}{n}\right) + \left(\sum_{s=0}^{t-1} R_s W^{t-s-1}e_i - \frac{R_s\mathbf{1}_n}{n}\right)\right\|^2\right]$$

$$= 2\sum_{i=1}^n \mathbb{E}_t\left[\left\|\eta\left(\sum_{s=0}^{t-1} G(X_s)W^{t-s-1}e_i - \frac{G(X_s)\mathbf{1}_n}{n}\right)\right\|^2\right]$$

$$+ 2\sum_{i=1}^n \mathbb{E}_t\left[\left\|\left(\sum_{s=0}^{t-1} M_s W^{t-s-1}e_i - \frac{M_s\mathbf{1}_n}{n}\right)\right\|^2\right]$$

$$+ 2\sum_{i=1}^n \mathbb{E}_t\left[\left\|\left(\sum_{s=0}^{t-1} R_s W^{t-s-1}e_i - \frac{R_s\mathbf{1}_n}{n}\right)\right\|^2\right]$$

$$+ 4\sum_{i=1}^n \mathbb{E}_t\left[\left\langle\sum_{s=0}^{t-1}\mathbb{E}_t\left[M_s\right]\left(W^{t-s-1}e_i - \frac{\mathbf{1}_n}{n}\right), \sum_{s=0}^{t-1}\mathbb{E}_t\left[R_s\right]\left(W^{t-s-1}e_i - \frac{\mathbf{1}_n}{n}\right)\right\rangle\right]$$

$$= 2\sum_{i=1}^n \mathbb{E}_t\left[\left\|\eta\left(\sum_{s=0}^{t-1} G(X_s)W^{t-s-1}e_i - \frac{G(X_s)\mathbf{1}_n}{n}\right)\right\|^2\right]$$

$$+ 2\sum_{i=1}^n \mathbb{E}_t\left[\left\|\left(\sum_{s=0}^{t-1} M_s W^{t-s-1}e_i - \frac{M_s\mathbf{1}_n}{n}\right)\right\|^2\right]$$

$$+ 2\sum_{i=1}^n \mathbb{E}_t\left[\left\|\left(\sum_{s=0}^{t-1} R_s W^{t-s-1}e_i - \frac{R_s\mathbf{1}_n}{n}\right)\right\|^2\right]$$

$$= 2\sum_{i=1}^n \mathbb{E}_t\left[\left\|\eta\left(\sum_{s=0}^{t-1} G(X_s)W^{t-s-1}e_i - \frac{G(X_s)\mathbf{1}_n}{n}\right)\right\|^2\right]$$

$$+ 2\sum_{i=1}^{n}\sum_{s=0}^{t-1}\mathbb{E}_t\left[\left\|\left(M_s W^{t-s-1} e_i - \frac{M_s \mathbf{1}_n}{n}\right)\right\|^2\right]$$

$$+ 2\sum_{i=1}^{n}\sum_{s=0}^{t-1}\mathbb{E}_t\left[\left\|\left(R_s W^{t-s-1} e_i - \frac{R_s \mathbf{1}_n}{n}\right)\right\|^2\right]$$

$$+ 4\sum_{i=1}^{n}\sum_{s\neq s'}\mathbb{E}_t\left[\left\langle \mathbb{E}_t\left[M_s\right]\left(W^{t-s-1}e_i - \frac{\mathbf{1}_n}{n}\right), \mathbb{E}_t\left[M_{s'}\right]\left(W^{t-s'-1}e_i - \frac{\mathbf{1}_n}{n}\right)\right\rangle\right]$$

$$+ 4\sum_{i=1}^{n}\sum_{s\neq s'}\mathbb{E}_t\left[\left\langle \mathbb{E}_t\left[R_s\right]\left(W^{t-s-1}e_i - \frac{\mathbf{1}_n}{n}\right), \mathbb{E}_t\left[R_{s'}\right]\left(W^{t-s'-1}e_i - \frac{\mathbf{1}_n}{n}\right)\right\rangle\right]$$

$$= 2\sum_{i=1}^{n}\mathbb{E}_t\left[\left\|\eta\left(\sum_{s=0}^{t-1}G(X_s)\left(W^{t-s-1} - \frac{\mathbf{1}_n\mathbf{1}_n^{\top}}{n}\right)e_i\right)\right\|^2\right]$$

$$+ 2\sum_{i=1}^{n}\sum_{s=0}^{t-1}\mathbb{E}_t\left[\left\|\left(M_s\left(W^{t-s-1} - \frac{\mathbf{1}_n\mathbf{1}_n^{\top}}{n}\right)e_i\right)\right\|^2\right]$$

$$+ 2\sum_{i=1}^{n}\sum_{s=0}^{t-1}\mathbb{E}_t\left[\left\|\left(R_s\left(W^{t-s-1} - \frac{\mathbf{1}_n\mathbf{1}_n^{\top}}{n}\right)e_i\right)\right\|^2\right]$$

$$= 2\mathbb{E}_t\left[\left\|\eta\left(\sum_{s=0}^{t-1}G(X_s)\left(W^{t-s-1} - v_1 v_1^{\top}\right)\right)\right\|_F^2\right]$$

$$+ 2\sum_{s=0}^{t-1}\mathbb{E}_t\left[\left\|\left(M_s\left(W^{t-s-1} - v_1 v_1^{\top}\right)\right)\right\|_F^2\right]$$

$$+ 2\sum_{s=0}^{t-1}\mathbb{E}_t\left[\left\|\left(R_s\left(W^{t-s-1} - v_1 v_1^{\top}\right)\right)\right\|_F^2\right]$$

$$= 2\mathbb{E}_t\left[\left\|\eta\left(\sum_{s=0}^{t-1}G(X_s)\left(P\Lambda^{t-s-1}P^{\top} - v_1 v_1^{\top}\right)\right)\right\|_F^2\right]$$

$$+ 2\sum_{s=0}^{t-1}\mathbb{E}_t\left[\left\|\left(M_s\left(P\Lambda^{t-s-1}P^{\top} - v_1 v_1^{\top}\right)\right)\right\|_F^2\right]$$

$$+ 2\sum_{s=0}^{t-1}\mathbb{E}_t\left[\left\|\left(R_s\left(P\Lambda^{t-s-1}P^{\top} - v_1 v_1^{\top}\right)\right)\right\|_F^2\right]$$

$$= 2\mathbb{E}_t\left[\left\|\eta\sum_{s=0}^{t-1}G(X_s)P\left(\begin{bmatrix}1 & & & \\ & \lambda_2^{t-s-1} & & \\ & & \ddots & \\ & & & \lambda_n^{t-s-1}\end{bmatrix} - \begin{bmatrix}1 & & & \\ & 0 & & \\ & & \ddots & \\ & & & 0\end{bmatrix}\right)P^{\top}\right\|_F^2\right]$$

$$+ 2\sum_{s=0}^{t-1}\mathbb{E}_t\left[\left\|M_s P\left(\begin{bmatrix}1 & & & \\ & \lambda_2^{t-s-1} & & \\ & & \ddots & \\ & & & \lambda_n^{t-s-1}\end{bmatrix} - \begin{bmatrix}1 & & & \\ & 0 & & \\ & & \ddots & \\ & & & 0\end{bmatrix}\right)P^{\top}\right\|_F^2\right]$$

$$+ 2\sum_{s=0}^{t-1}\mathbb{E}_t\left[\left\|R_s P\left(\begin{bmatrix}1 & & & \\ & \lambda_2^{t-s-1} & & \\ & & \ddots & \\ & & & \lambda_n^{t-s-1}\end{bmatrix} - \begin{bmatrix}1 & & & \\ & 0 & & \\ & & \ddots & \\ & & & 0\end{bmatrix}\right)P^{\top}\right\|_F^2\right]$$

$$= 2\mathbb{E}_t \left[ \left\| \eta \sum_{s=0}^{t-1} G(X_s)P \left( \begin{bmatrix} 0 & & & \\ & \lambda_2^{t-s-1} & & \\ & & \ddots & \\ & & & \lambda_n^{t-s-1} \end{bmatrix} \right) P^\top \right\|_F^2 \right]$$

$$+ 2\sum_{s=0}^{t-1} \mathbb{E}_t \left[ \left\| M_s P \left( \begin{bmatrix} 0 & & & \\ & \lambda_2^{t-s-1} & & \\ & & \ddots & \\ & & & \lambda_n^{t-s-1} \end{bmatrix} \right) P^\top \right\|_F^2 \right]$$

$$+ 2\sum_{s=0}^{t-1} \mathbb{E}_t \left[ \left\| R_s P \left( \begin{bmatrix} 0 & & & \\ & \lambda_2^{t-s-1} & & \\ & & \ddots & \\ & & & \lambda_n^{t-s-1} \end{bmatrix} \right) P^\top \right\|_F^2 \right]$$

$$= 2\mathbb{E}_t \left[ \left\| \eta \sum_{s=0}^{t-1} G(X_s)P \left( \begin{bmatrix} 0 & & & \\ & \lambda_2^{t-s-1} & & \\ & & \ddots & \\ & & & \lambda_n^{t-s-1} \end{bmatrix} \right) \right\|_F^2 \right]$$

$$+ 2\sum_{s=0}^{t-1} \mathbb{E}_t \left[ \left\| M_s P \left( \begin{bmatrix} 0 & & & \\ & \lambda_2^{t-s-1} & & \\ & & \ddots & \\ & & & \lambda_n^{t-s-1} \end{bmatrix} \right) \right\|_F^2 \right]$$

$$+ 2\sum_{s=0}^{t-1} \mathbb{E}_t \left[ \left\| R_s P \left( \begin{bmatrix} 0 & & & \\ & \lambda_2^{t-s-1} & & \\ & & \ddots & \\ & & & \lambda_n^{t-s-1} \end{bmatrix} \right) \right\|_F^2 \right]$$

$$\leq 2\mathbb{E}_t \left[ \left\| \eta \sum_{s=0}^{t-1} \rho^{t-s-1} G(X_s)P \right\|_F^2 \right] + 2\sum_{s=0}^{t-1} \mathbb{E}_t \left[ \left\| \rho^{t-s-1} M_s P \right\|_F^2 \right] + 2\sum_{s=0}^{t-1} \mathbb{E}_t \left[ \left\| \rho^{t-s-1} R_s P \right\|_F^2 \right]$$

$$= 2\mathbb{E}_t \left[ \left( \sum_{s=0}^{t-1} \eta \rho^{t-s-1} \left\| G(X_s) \right\|_F \right)^2 \right] + 2\sum_{s=0}^{t-1} \mathbb{E}_t \left[ \left\| \rho^{t-s-1} M_s \right\|_F^2 \right] + 2\sum_{s=0}^{t-1} \mathbb{E}_t \left[ \left\| \rho^{t-s-1} R_s \right\|_F^2 \right],$$

$$\tag{B.10}$$

where the (a) in (B.10) applies the update rule in (B.9), the $X_0$ term is removed since all nodes share the same initial model parameters $X_0$.

Therefore, after summing over $T - 1$ iterations, for the third term in (B.10), using the fact that the initial parameters are same when $t = 0$ and the initial value $R_{-1} = 0$, we have

$$\sum_{t=0}^{T-1} \sum_{s=0}^{t-1} \mathbb{E}_t \left[ \left\| \rho^{t-s-1} R_s \right\|_F^2 \right] = \sum_{t=1}^{T-1} \sum_{s=0}^{t-1} \mathbb{E}_t \left[ \left\| \rho^{t-s-1} R_s \right\|_F^2 \right] = \sum_{s=0}^{T-2} \sum_{t=s+1}^{T-1} \rho^{2(t-s-1)} \mathbb{E}_t \left[ \left\| R_s \right\|_F^2 \right]$$

$$= \sum_{s=0}^{T-2} \sum_{t'=0}^{T-s-2} \rho^{2t'} \mathbb{E}_t \left[ \left\| R_s \right\|_F^2 \right] \overset{(a)}{\leq} \frac{1}{1-\rho^2} \sum_{s=0}^{T-2} \mathbb{E}_t \left[ \left\| R_s \right\|_F^2 \right] \leq \frac{1}{1-\rho^2} \sum_{s=0}^{T-1} \mathbb{E}_t \left[ \left\| R_s \right\|_F^2 \right], \quad \text{(B.11)}$$

where (a) is due to $\sum_{t'=0}^{T-s-2} \rho^{2t'} \leq \sum_{t'=0}^{\infty} \rho^{2t'} = \frac{1}{1-\rho^2}$. The same technique can be applied to the second term in (B.10).

Let $G(X_{-1}) = 0$, the first term in (B.10) can be computed as

$$\sum_{t=0}^{T-1} \mathbb{E}_t \left[ \left( \sum_{s=0}^{t-1} \eta \rho^{t-s-1} \left\| G(X_s) \right\|_F \right)^2 \right]$$

$$= \sum_{t=1}^{T-1} \mathbb{E}_t \left[ \left( \sum_{s=0}^{t-1} \eta \rho^{t-s-1} \left\| G(X_s) \right\|_F \right)^2 \right]$$

$$= \eta^2 \sum_{t=1}^{T-1} \sum_{s=0}^{t-1} \sum_{s'=0}^{t-1} \rho^{t-s-1} \rho^{t-s'-1} \mathbb{E}_t \left[ \left\| G(X_s) \right\|_F \right] \mathbb{E}_t \left[ \left\| G(X_{s'}) \right\|_F \right]$$

$$\overset{(a)}{\leq} \eta^2 \sum_{t=1}^{T-1} \sum_{s=0}^{t-1} \sum_{s'=0}^{t-1} \rho^{t-s-1} \rho^{t-s'-1} \left( \frac{\mathbb{E}_t \left[ \left\| G(X_s) \right\|_F \right]^2 + \mathbb{E}_t \left[ \left\| G(X_{s'}) \right\|_F \right]^2}{2} \right)$$

$$= \frac{\eta^2}{2} \sum_{t=1}^{T-1} \sum_{s=0}^{t-1} \rho^{t-s-1} \mathbb{E}_t \left[ \left\| G(X_s) \right\|_F^2 \right] \sum_{s'=0}^{t-1} \rho^{t-s'-1}$$

$$+ \frac{\eta^2}{2} \sum_{t=1}^{T-1} \sum_{s'=0}^{t-1} \rho^{t-s'-1} \mathbb{E}_t \left[ \left\| G(X_{s'}) \right\|_F^2 \right] \sum_{s=0}^{t-1} \rho^{t-s-1}$$

$$= \frac{\eta^2}{2} \sum_{t=1}^{T-1} \sum_{s=0}^{t-1} \rho^{t-s-1} \mathbb{E}_t \left[ \left\| G(X_s) \right\|_F^2 \right] \sum_{r=0}^{t-1} \rho^r + \frac{\eta^2}{2} \sum_{t=1}^{T-1} \sum_{s'=0}^{t-1} \rho^{t-s'-1} \mathbb{E}_t \left[ \left\| G(X_{s'}) \right\|_F^2 \right] \sum_{r'=0}^{t-1} \rho^{r'}$$

$$\leq \frac{\eta^2}{2(1-\rho)} \sum_{t=1}^{T-1} \sum_{s=0}^{t-1} \rho^{t-s-1} \mathbb{E}_t \left[ \left\| G(X_s) \right\|_F^2 \right] + \frac{\eta^2}{2(1-\rho)} \sum_{t=1}^{T-1} \sum_{s'=0}^{t-1} \rho^{t-s'-1} \mathbb{E}_t \left[ \left\| G(X_{s'}) \right\|_F^2 \right]$$

$$\leq \frac{\eta^2}{2(1-\rho)^2} \sum_{s=0}^{T-2} \mathbb{E}_t \left[ \left\| G(X_s) \right\|_F^2 \right] + \frac{\eta^2}{2(1-\rho)^2} \sum_{s'=0}^{T-2} \mathbb{E}_t \left[ \left\| G(X_{s'}) \right\|_F^2 \right]$$

$$= \frac{\eta^2}{(1-\rho)^2} \sum_{s=0}^{T-2} \mathbb{E}_t \left[ \left\| G(X_s) \right\|_F^2 \right]$$

$$\leq \frac{\eta^2}{(1-\rho)^2} \sum_{s=0}^{T-1} \mathbb{E}_t \left[ \left\| G(X_s) \right\|_F^2 \right], \tag{B.12}$$

where we apply $2ab \leq a^2 + b^2$ in (a).

Then, we plug (B.11) and (B.12) back into (B.10), and we have

$$\sum_{t=0}^{T-1} \sum_{i=1}^{n} \mathbb{E}_t \left[ \left\| \frac{X_t \mathbf{1}_n}{n} - x_t^i \right\|^2 \right] \leq \frac{2}{1-\rho^2} \sum_{t=0}^{T-1} \mathbb{E}_t \left[ \left\| M_t \right\|_F^2 \right] + \frac{2}{1-\rho^2} \sum_{t=0}^{T-1} \mathbb{E}_t \left[ \left\| R_t \right\|_F^2 \right]$$

$$+ \frac{2\eta^2}{(1-\rho)^2} \sum_{t=0}^{T-1} \mathbb{E}_t \left[ \left\| G(X_t) \right\|_F^2 \right]. \tag{B.13}$$

Let $H_t = \left[ e_t^1, \ldots, e_t^n \right]$, then, inequality (B.8) becomes

$$\frac{1}{T} \sum_{t=0}^{T-1} \mathbb{E}_t \left[ f \left( \frac{X_{t+1} \mathbf{1}_n}{n} \right) \right] \leq \frac{1}{T} \sum_{t=0}^{T-1} \mathbb{E}_t \left[ f \left( \frac{X_t \mathbf{1}_n}{n} \right) \right] - \frac{\eta}{2} \frac{1}{T} \sum_{t=0}^{T-1} \mathbb{E}_t \left[ \left\| \nabla f \left( \frac{X_t \mathbf{1}_n}{n} \right) \right\|^2 \right]$$

$$- \frac{\eta}{2} \frac{1}{T} \sum_{t=0}^{T-1} \mathbb{E}_t \left[ \left\| \overline{\nabla f} \left( X_t \right) \right\|^2 \right] + \frac{\eta^2 L}{2} \frac{1}{T} \sum_{t=0}^{T-1} \mathbb{E}_t \left[ \left\| \overline{\nabla f} \left( X_t \right) \right\|^2 \right]$$

$$+ \frac{\eta^2 \sigma^2 L}{2n} + \frac{\eta^3 L^2}{n(1-\rho)^2 T} \sum_{t=0}^{T-1} \mathbb{E}_t \left[ \|G(X_t)\|_F^2 \right]$$

$$+ \left( \frac{\eta\gamma^2 L^2}{n(1-\rho^2)} + \frac{\gamma^2 L}{2n^2} \right) \frac{1}{T} \sum_{t=0}^{T-1} \mathbb{E}_t \left[ \|H_t\|_F^2 \right]$$

$$+ \left( \frac{\eta L^2}{n(1-\rho^2)} + \frac{L}{2n^2} \right) \frac{1}{T} \sum_{t=0}^{T-1} \mathbb{E}_t \left[ \|R_t\|_F^2 \right]. \tag{B.14}$$

Before further simplification, we provide the following expression.

$$X_{t+1} = X_t W - \eta G(X_t) + M_t + R_t = X_t P \Lambda P^\top - \eta G(X_t) + M_t + R_t, \tag{B.15}$$

$$X_{t+1} P = X_t P \Lambda - \eta G(X_t) P + M_t P + R_t P, \tag{B.16}$$

and

$$Y_{t+1} = Y_t \Lambda - \eta J_t + N_t + K_t. \tag{B.17}$$

So $\sum_{t=0}^{T-1} \mathbb{E}_t \left[ \|X_t (W - I)\|_F^2 \right] = \sum_{t=0}^{T-1} \mathbb{E}_t \left[ \|X_t P (\Lambda - I) P^\top\|_F^2 \right] = \sum_{t=0}^{T-1} \mathbb{E}_t \left[ \|Y_t (\Lambda - I)\|_F^2 \right]$. We also know connected matrix $W$ is a doubly symmetric stochastic matrix and the formula for eigen-decomposition is $W = P\Lambda P^\top$, where $P = (v_1, v_2, \ldots, v_n)$ and $P^\top P = PP^\top = I$. So, we have $WP = P\Lambda$ and $Y_0 \Lambda = X_0 P \Lambda = X_0 W P = X_0 P = Y_0$, then according to the update rule of $Y_t$, we can rewrite $Y_t (\Lambda - I) = Y_t \Lambda - Y_t$ as follows.

$$Y_t (\Lambda - I) = Y_t \Lambda - Y_t = \left( \sum_{s=0}^{t-1} \Lambda^{t-s-1} (-\eta J_s + N_s + K_s) \right) (\Lambda - I). \tag{B.18}$$

Note that we have

$$\left\| b_t^i - C(b_t^i) \right\|^2 \leq \beta \left\| b_t^i \right\|^2,$$

and apply inequality

$$\|x_1 + x_2\|^2 \leq (1 + c_1) \|x_1\|^2 + (1 + c_1^{-1}) \|x_2\|^2,$$

where $c_1$ is a constant which satisfies $c_1 > 0$.

Then, we have

$$\sum_{t=0}^{T-1} \mathbb{E}_t \left[ \|R_t\|_F^2 \right]$$

$$\leq \beta \sum_{t=0}^{T-1} \mathbb{E}_t \left[ \|X_t(W - I) - \eta G(X_t) + \gamma H_t\|_F^2 \right]$$

$$\leq \beta(1 + a) \sum_{t=0}^{T-1} \mathbb{E}_t \left[ \|X_t(W - I)\|_F^2 \right] + \beta(1 + a^{-1}) \sum_{t=0}^{T-1} \mathbb{E}_t \left[ \|-\eta G(X_t) + \gamma H_t\|_F^2 \right]$$

$$\leq \beta(1 + a) \sum_{t=0}^{T-1} \mathbb{E}_t \left[ \|X_t(W - I)\|_F^2 \right] + 2\beta\eta^2(1 + a^{-1}) \sum_{t=0}^{T-1} \mathbb{E}_t \left[ \|G(X_t)\|_F^2 \right]$$

$$+ 2\beta\gamma^2(1 + a^{-1}) \sum_{t=0}^{T-1} \mathbb{E}_t\left[\|H_t\|_F^2\right]$$

$$\overset{(a)}{\leq} \frac{\beta\mu^2(1+a)}{(1-\rho)^2} \sum_{t=0}^{T-1} \mathbb{E}_t\left[\|-\eta J_t + K_t + N_t\|_F^2\right]$$

$$+ 2\beta\eta^2(1 + a^{-1}) \sum_{t=0}^{T-1} \mathbb{E}_t\left[\|G(X_t)\|_F^2\right] + 2\beta\gamma^2(1 + a^{-1}) \sum_{t=0}^{T-1} \mathbb{E}_t\left[\|H_t\|_F^2\right]$$

$$\leq \frac{\beta\mu^2(1+a)(1+b)}{(1-\rho)^2} \sum_{t=0}^{T-1} \mathbb{E}_t\left[\|K_t\|_F^2\right] + \frac{\beta\mu^2(1+a)(1+b^{-1})}{(1-\rho)^2} \sum_{t=0}^{T-1} \mathbb{E}_t\left[\|-\eta J_t + N_t\|_F^2\right]$$

$$+ 2\beta\eta^2(1 + a^{-1}) \sum_{t=0}^{T-1} \mathbb{E}_t\left[\|G(X_t)\|_F^2\right] + 2\beta\gamma^2(1 + a^{-1}) \sum_{t=0}^{T-1} \mathbb{E}_t\left[\|H_t\|_F^2\right]$$

$$\leq \frac{\beta\mu^2(1+a)(1+b)}{(1-\rho)^2} \sum_{t=0}^{T-1} \mathbb{E}_t\left[\|K_t\|_F^2\right] + \frac{\beta\mu^2\eta^2(1+a)(1+b^{-1})(1+c)}{(1-\rho)^2} \sum_{t=0}^{T-1} \mathbb{E}_t\left[\|J_t\|_F^2\right]$$

$$+ \frac{\beta\mu^2(1+a)(1+b^{-1})(1+c^{-1})}{(1-\rho)^2} \sum_{t=0}^{T-1} \mathbb{E}_t\left[\|N_t\|_F^2\right] + 2\beta\eta^2(1 + a^{-1}) \sum_{t=0}^{T-1} \mathbb{E}_t\left[\|G(X_t)\|_F^2\right]$$

$$+ 2\beta\gamma^2(1 + a^{-1}) \sum_{t=0}^{T-1} \mathbb{E}_t\left[\|H_t\|_F^2\right]$$

$$\leq \frac{\beta\mu^2(1+a)(1+b)}{(1-\rho)^2} \sum_{t=0}^{T-1} \mathbb{E}_t\left[\|R_t\|_F^2\right] + \frac{\beta\mu^2\eta^2(1+a)(1+b^{-1})(1+c)}{(1-\rho)^2} \sum_{t=0}^{T-1} \mathbb{E}_t\left[\|G(X_t)\|_F^2\right]$$

$$+ \frac{\beta\mu^2\gamma^2(1+a)(1+b^{-1})(1+c^{-1})}{(1-\rho)^2} \sum_{t=0}^{T-1} \mathbb{E}_t\left[\|H_t\|_F^2\right] + 2\beta\eta^2(1 + a^{-1}) \sum_{t=0}^{T-1} \mathbb{E}_t\left[\|G(X_t)\|_F^2\right]$$

$$+ 2\beta\gamma^2(1 + a^{-1}) \sum_{t=0}^{T-1} \mathbb{E}_t\left[\|H_t\|_F^2\right], \tag{B.19}$$

where $(a)$ applies the following technique. We further define $\mu = \max_{i=2,3,\dots,n} |\lambda_i - 1|$ and obtain

$$\beta(1+a) \sum_{t=0}^{T-1} \mathbb{E}_t\left[\|X_t(W - I)\|_F^2\right] = \beta(1+a) \sum_{t=0}^{T-1} \mathbb{E}_t\left[\|Y_t(\Lambda - I)\|_F^2\right]$$

$$= \beta(1+a) \sum_{t=0}^{T-1} \mathbb{E}_t\left[\left\|\left(\sum_{s=0}^{t-1} \Lambda^{t-s-1}(-\eta J_s + N_s + K_s)\right)(\Lambda - I)\right\|_F^2\right]$$

$$= \beta(1+a) \sum_{t=0}^{T-1} \mathbb{E}_t\left[\left\|\left(\sum_{s=0}^{t-1} \Lambda^{t-s-1}(-\eta J_s + N_s + K_s)\right)\left(\begin{bmatrix} 0 & & & \\ & \lambda_2 - 1 & & \\ & & \ddots & \\ & & & \lambda_n - 1 \end{bmatrix}\right)\right\|_F^2\right]$$

$$\leq \beta\mu^2(1+a) \sum_{t=0}^{T-1} \sum_{i=2}^{n} \mathbb{E}_t\left[\left\|\sum_{s=0}^{t-1} \lambda_i^{t-s-1}(-\eta j_s^i + n_s^i + k_s^i)\right\|^2\right]$$

$$= \beta\mu^2(1+a) \sum_{t=0}^{T-1} \sum_{i=2}^{n} \mathbb{E}_t\left[\left(\sum_{s=0}^{t-1} |\lambda_i^{t-s-1}| \, \|-\eta j_s^i + n_s^i + k_s^i\|\right)^2\right]$$

$$\leq \beta\mu^2(1+a)\sum_{t=0}^{T-1}\sum_{i=2}^{n}\mathbb{E}_t\left[\left(\sum_{s=0}^{t-1}\rho^{t-s-1}\left\|-\eta j_s^i + n_s^i + k_s^i\right\|\right)^2\right]$$

$$\overset{(a)}{\leq}\frac{\beta\mu^2(1+a)}{(1-\rho)^2}\sum_{t=0}^{T-1}\sum_{i=2}^{n}\mathbb{E}_t\left[\left\|-\eta j_t^i + n_t^i + k_t^i\right\|^2\right]$$

$$\leq \frac{\beta\mu^2(1+a)}{(1-\rho)^2}\sum_{t=0}^{T-1}\mathbb{E}_t\left[\left\|-\eta J_t + N_t + K_t\right\|_F^2\right], \tag{B.20}$$

where (a) applies the same technique in (B.12).

By rearranging (B.19), we have the following expression for $\|R_t\|_F$.

$$\sum_{t=0}^{T-1}\mathbb{E}_t\left[\|R_t\|_F^2\right] \leq \frac{\beta\gamma^2(\mu^2(1+a)(1+b^{-1})(1+c^{-1}) + 2(1+a^{-1})(1-\rho)^2)}{(1-\rho)^2 - \beta\mu^2(1+a)(1+b)}\sum_{t=0}^{T-1}\mathbb{E}_t\left[\|H_t\|_F^2\right]$$

$$+ \frac{\beta\eta^2(\mu^2(1+a)(1+b^{-1})(1+c) + 2(1+a^{-1})(1-\rho)^2)}{(1-\rho)^2 - \beta\mu^2(1+a)(1+b)}\sum_{t=0}^{T-1}\mathbb{E}_t\left[\|G(X_t)\|_F^2\right]. \tag{B.21}$$

We know that the Frobenius Norm $\|A\|_F^2 = \sum_{i=1}^{n}\|a_i\|^2$, we have $\sum_{t=0}^{T-1}\sum_{i=1}^{n}\mathbb{E}_t\left[\left\|-e_{t+1}^i\right\|^2\right] = \sum_{t=0}^{T-1}\sum_{i=1}^{n}\mathbb{E}_t\left[\left\|e_{t+1}^i\right\|^2\right] = \sum_{t=0}^{T-1}\sum_{i=1}^{n}\mathbb{E}_t\left[\left\|r_t^i\right\|^2\right] = \sum_{t=0}^{T-1}\mathbb{E}_t\left[\|R_t\|_F^2\right]$. Now, we have

$$\sum_{t=0}^{T-1}\sum_{i=1}^{n}\mathbb{E}_t\left[\left\|e_{t+1}^i\right\|^2\right] \leq A_1\sum_{t=0}^{T-1}\sum_{i=1}^{n}\mathbb{E}_t\left[\left\|e_t^i\right\|^2\right] + A_2\sum_{t=0}^{T-1}\sum_{i=1}^{n}\mathbb{E}_t\left[\left\|G_i(X_t^i)\right\|^2\right], \tag{B.22}$$

where

$$A_1 = \frac{\beta\gamma^2(\mu^2(1+a)(1+b^{-1})(1+c^{-1}) + 2(1+a^{-1})(1-\rho)^2)}{(1-\rho)^2 - \beta\mu^2(1+a)(1+b)}, \tag{B.23}$$

and

$$A_2 = \frac{\beta\eta^2(\mu^2(1+a)(1+b^{-1})(1+c) + 2(1+a^{-1})(1-\rho)^2)}{(1-\rho)^2 - \beta\mu^2(1+a)(1+b)}. \tag{B.24}$$

Then, we have the expression of $\mathbb{E}_t\left[\left\|e_t^i\right\|^2\right]$ as follows.

$$\mathbb{E}_t\left[\left\|e_t^i\right\|^2\right] \leq A_1^t\mathbb{E}_t\left[\left\|e_0^i\right\|^2\right] + A_2\sum_{s=0}^{t-1}A_1^{t-s-1}\mathbb{E}_t\left[\left\|G_i(X_s^i)\right\|^2\right]. \tag{B.25}$$

We know that $e_0^i = 0$. Therefore, by applying the geometric series, we have

$$\sum_{t=0}^{T-1}\sum_{i=1}^{n}\mathbb{E}_t\left[\left\|e_t^i\right\|^2\right]$$

$$\leq A_2\sum_{i=1}^{n}\sum_{t=0}^{T-1}\sum_{s=0}^{t-1}A_1^{t-s-1}\mathbb{E}_t\left[\left\|G_i(X_s^i)\right\|^2\right] = A_2\sum_{i=1}^{n}\sum_{s=0}^{T-2}\sum_{t=0}^{T-1}A_1^{t-s-1}\mathbb{E}_t\left[\left\|G_i(X_s^i)\right\|^2\right]$$

$$= A_2\sum_{i=1}^{n}\sum_{s=0}^{T-2}\sum_{t'=0}^{T-2}A_1^{t'}\mathbb{E}_t\left[\left\|G_i(X_s^i)\right\|^2\right] \leq \frac{A_2}{1-A_1}\sum_{t=0}^{T-2}\sum_{i=1}^{n}\mathbb{E}_t\left[\left\|G_i(X_t^i)\right\|^2\right]$$

$$\leq \frac{A_2}{1-A_1}\sum_{t=0}^{T-1}\sum_{i=1}^{n}\mathbb{E}_t\left[\left\|G_i(X_t^i)\right\|^2\right] = C_1\sum_{t=0}^{T-1}\sum_{i=1}^{n}\mathbb{E}_t\left[\left\|G_i(X_t^i)\right\|^2\right], \tag{B.26}$$

where we define $C_1$ as

$$C_1 = \frac{\beta\mu^2\eta^2(1+a)(1+b^{-1})(1+c) + 2\beta\eta^2(1+a^{-1})(1-\rho)^2}{(1-\rho)^2 - \beta(\mu^2(1+a)(1+b) + \mu^2\gamma^2(1+a)(1+b^{-1})(1+c^{-1}) + 2\gamma^2(1+a^{-1})(1-\rho)^2)}.$$

(B.27)

Then, combining (B.26) and (B.22), we have

$$\sum_{t=0}^{T-1} \mathbb{E}_t\left[\|H_t\|_F^2\right] \leq C_1 \sum_{t=0}^{T-1} \mathbb{E}_t\left[\|G(X_t)\|_F^2\right].$$

(B.28)

Then, combining (B.28) and (B.21), we have

$$\sum_{t=0}^{T-1} \mathbb{E}_t\left[\|R_t\|_F^2\right] \leq C_1 \sum_{t=0}^{T-1} \mathbb{E}_t\left[\|G(X_t)\|_F^2\right].$$

(B.29)

Next, we compute $\sum_{t=0}^{T-1}\sum_{i=1}^{n}\mathbb{E}_t\left[\left\|G_i(x_t^i)\right\|^2\right]$ as follows.

$$\sum_{t=0}^{T-1}\sum_{i=1}^{n}\mathbb{E}_t\left[\left\|G_i(x_t^i)\right\|^2\right] = \sum_{t=0}^{T-1}\sum_{i=1}^{n}\mathbb{E}_t\left[\left\|\nabla F_i\left(x_t^i;\xi_t^i\right)\right\|^2\right]$$

$$= \sum_{t=0}^{T-1}\sum_{i=1}^{n}\mathbb{E}_t\left[\left\|\nabla F_i\left(x_t^i;\xi_t^i\right) - \overline{\nabla f}\left(X_t\right) + \overline{\nabla f}\left(X_t\right)\right\|^2\right]$$

$$\leq \sum_{t=0}^{T-1}\sum_{i=1}^{n}2\mathbb{E}_t\left[\left\|\overline{\nabla f}\left(X_t\right)\right\|^2\right] + \sum_{t=0}^{T-1}\sum_{i=1}^{n}2\mathbb{E}_t\left[\left\|\nabla F_i\left(x_t^i;\xi_t^i\right) - \overline{\nabla f}\left(X_t\right)\right\|^2\right]$$

$$= \sum_{t=0}^{T-1}2n\mathbb{E}_t\left[\left\|\overline{\nabla f}\left(X_t\right)\right\|^2\right]$$

$$+ \sum_{t=0}^{T-1}\sum_{i=1}^{n}2\mathbb{E}_t\left[\left\|\nabla F_i\left(x_t^i;\xi_t^i\right) - \nabla f_i\left(x_t^i\right) + \nabla f_i\left(x_t^i\right) - \overline{\nabla f}\left(X_t\right)\right\|^2\right]$$

$$\leq \sum_{t=0}^{T-1}2n\mathbb{E}_t\left[\left\|\overline{\nabla f}\left(X_t\right)\right\|^2\right]$$

$$+ \sum_{t=0}^{T-1}\sum_{i=1}^{n}4\mathbb{E}_t\left[\left\|\nabla F_i\left(x_t^i;\xi_t^i\right) - \nabla f_i\left(x_t^i\right)\right\|^2\right]$$

$$+ \sum_{t=0}^{T-1}\sum_{i=1}^{n}4\mathbb{E}_t\left[\left\|\nabla f_i\left(x_t^i\right) - \overline{\nabla f}\left(X_t\right)\right\|^2\right]$$

$$\leq \sum_{t=0}^{T-1}2n\mathbb{E}_t\left[\left\|\overline{\nabla f}\left(X_t\right)\right\|^2\right] + 4\left(\sigma^2 + \epsilon^2\right)nT.$$

(B.30)

Combining (B.14), (B.28), (B.29) and (B.30), and taking the total expectation and rearranging, we have

$$\frac{1}{T}\sum_{t=0}^{T-1}\left(\mathbb{E}\left[\nabla f\left(\frac{X_t\mathbf{1}_n}{n}\right)\right] + (1-B_1)\mathbb{E}\left[\overline{\nabla f}\left(X_t\right)\right]\right)$$

$$\leq \frac{2(f(0) - f^*)}{\eta T}$$

$$+ \left(\frac{\eta L}{n} + \frac{8\eta^2 L^2}{(1-\rho)^2} + 8\eta C_1(1+\gamma^2)\left(\frac{\eta L^2}{1-\rho^2} + \frac{L}{2n}\right)\right)\sigma^2$$

$$+ \left( \frac{8\eta^2 L^2}{(1-\rho)^2} + 8\eta C_1 (1+\gamma^2) \left( \frac{\eta L^2}{1-\rho^2} + \frac{L}{2n} \right) \right) \epsilon^2, \tag{B.31}$$

where $B_1$ and $C_1$ are defined in Theorem 1, $f(0)$ is the initial model parameters which is the same for all the nodes and $f^*$ is the optimal solution for function $f$.

Hence, we finish the proof of Theorem 1.

