# OpenReview forum: "Compressed Decentralized Learning with Error-Feedback under Data Heterogeneity"
_ICLR.cc/2025/Conference — ICLR 2025 Conference Withdrawn Submission_

### Official Review · Reviewer_2PpH · 2024-11-01

**Soundness:** 1
**Presentation:** 2
**Contribution:** 1
**Rating:** 1
**Confidence:** 5

**Summary:**

The paper investigates compressed decentralized learning in the presence of data heterogeneity, revealing limitations in existing algorithms when both gradient compression and data heterogeneity are high. To address these issues, the authors propose the DEFD-PSGD algorithm, designed to handle substantial gradient compression and data heterogeneity effectively, while maintaining communication efficiency. The authors provide theoretical analyses and validate their approach through numerical experiments.

**Strengths:**

This paper identifies a key challenge in the existing literature: the inability to handle both a high degree of gradient compression and significant data heterogeneity simultaneously.

**Weaknesses:**

Overall, this paper is not well-written and has several significant limitations, detailed as follows:

1. The proposed algorithm is impractical due to its high memory requirements. Specifically, the algorithm requires each node to store the model weights $\\{x^{i,j}\\}_{j \in \mathcal{N}_i}$ for all its neighbors, as seen on the left-hand side of line 4 and the right-hand side of line 7 in Algorithm 1. This poses a significant limitation. With large models reaching billions of parameters, even a single set of model weights requires substantial memory. For instance, storing one copy of a 175-billion-parameter model demands over 350GB, making the storage of neighbors' weights prohibitively memory-intensive and leading to notable hardware inefficiencies. Most existing compressed decentralized algorithms do not require each node to store neighbors' weights; see baseline algorithms like DCD-PSGD (Tang et al., 2018) and Choco-SGD (Koloskova et al., 2019). Moreover, other compressed decentralized methods, including those listed below [C1–C3], also avoid storing neighbors' weights. Thus, this requirement presents a strong limitation of the proposed algorithm.


2. While the authors claim that their convergence analysis is based on the most commonly used assumptions (see Contribution 2), this is unfortunately not the case. The authors assume that the compressors are both unbiased and contractive (i.e., $ \beta < 1 $; see the last two conditions in Assumption 1), which is rarely encountered in the literature. Typically, existing work assumes compressors are either unbiased—such as random sparsification [C4] and random quantization [C5]—or contractive, with top-$ K $ compression as an example, but do not assume them to hold simultaneously. Very few compressors satisfy both unbiasedness and contractiveness simultaneously. For instance, as shown in [C6], the random sparsification compressor [C4] is unbiased but has $ \beta = d/k - 1 $ where $ d \gg k $, and random quantization [C5] is unbiased with $ \beta > 2 $. Another common unbiased compressor, natural compression [C6], has $ \beta = 9/8 > 1 $. None of these widely used unbiased compressors meet the assumptions in this paper. Consequently, the assumptions here are restrictive and unlikely to hold in practical scenarios. This is another strong limitation. The standard assumption is to assume either unbiased or contractive compressors, but not both, see Assumption 3 and 4 in [C2].

3. The presented convergence results are weak. The convergence rate in (10) does not clarify the effects of network topology and the compression factor $\beta$. In contrast, existing literature, such as [C3], explicitly characterizes the influence of network topology and compression. Additionally, this paper appears unfamiliar with many state-of-the-art decentralized algorithms, including those listed in Table 1 of [C3]. Compared to these established algorithms, I do not observe clear advantages of the proposed approach's convergence rate. Moreover, it is important to note that the proposed algorithm relies on highly restrictive assumptions, such as simultaneous unbiasedness and contraction, along with bounded gradient dissimilarity. By comparison, [C2] and [C3] do not rely on these restrictive conditions.

4. The simulations are trivial. First of all, the tested top-K and the random quantization compressors do not satisfy the simulation unbiasedness and contraction property. Second, the baselines are trivial, more advanced baselines are in [C2, C3] as well as those listed in Table 1 of [C3]. Third, it is very strange that compressed algorithms on Cifar-10 can only achieve accuracy below 70%. The typical accuracy is above 90%.

[C1] Liu et. al., "LINEAR CONVERGENT DECENTRALIZED OPTIMIZATION WITH COMPRESSION", ICLR 2021

[C2] Huang and Pu, "CEDAS: A Compressed Decentralized Stochastic Gradient Method with Improved Convergence", arXiv 2301.05872, 2023

[C3] Islamov et.al., "Near Optimal Decentralized Optimization with Compression and Momentum Tracking", arXiv 2405.20114, 2024

[C4] Wangni et.al, "Gradient Sparsification for Communication-Efficient Distributed Optimization", NeurIPS 2018.

[C5] Alistarh et.al., "QSGD: Communication-Efficient SGD via Gradient Quantization and Encoding", NIPS 2017

[C6] Horváth et. al., "Natural Compression for Distributed Deep Learning", 2022

**Questions:**

1. Consider revising the algorithm design to eliminate the storage requirement for neighbors’ model weights. Existing algorithms avoid this need, and its inclusion here may render the algorithm impractical.

2. Please consider removing either the unbiasedness or the contractive assumption. Existing algorithms typically do not require both.

3. The coefficient $\beta$ is generally a fixed constant determined by the choice of compressor. For example, random sparsification yields $\beta = d/k - 1$, and random quantization results in $\beta > 1$. Why, then, do you assume that $\beta$ can be as small as you want? See your assumptions on the range of $\beta$ in Theorem 1, Corollary 1, and Corollary 2.

4. In Corollary 2, please clarify how network topology and the compression factor $\beta$ influence the convergence rate. Additionally, please compare your convergence rate with that of [C2, C3] and the algorithms listed in Table 1 of [C3], and explicitly outline the advantages of your algorithm.

5. Consider constructing compressors that simultaneously satisfy both the unbiased and contractive assumptions, and use these in your simulations. Additionally, please include more baseline comparisons beyond Choco-SGD and DCD-PSGD. Finally, clarify why your CIFAR-10 simulations yield low accuracy.

---

> ### Author Response · Authors · 2024-12-04
>
> Many thanks for your review. We appreciated some points you raised. For example, we agree with the fact that our paper was not very well written and can be significantly improved. We will revise our paper using the valuable points. Due to the extremely low and insulting score, we will withdraw our paper. However, we still would like to clarify some points that may be due to misunderstanding. In addition, we are very surprised to see the mistakes that the reviewer made in the review. It is also very surprising that the reviewer's confidence level is 5 given the fact that so many mistakes were made in the review. We felt very disappointed and upset by the quality of the review. Our responses to your comments are as follows.

---

> ### Author Response · Authors · 2024-12-04
>
> **Re: high memory requirements**
>
> First, we agree with the fact that our algorithm needs to store the neighbors' model weights. However, it is not uncommon in the existing literature that distributed training algorithms have a high memory requirement. For example, in [Tang et al.,2019], the DCD and ECD algorithms need the same memory requirement as our paper, and they also store the neighbors' model weights. In addition, the BEER algorithm in [1] and DoCoM in [2] mentioned by Reviewer k2Tp and MOTEF in [C3] have a higher memory requirement compared to our paper while they store the neighbors' model updates, which have the same number of parameters as the models themselves. The reviewer's statement that "Most existing compressed decentralized algorithms do not require each node to store neighbors' weights; see baseline algorithms like DCD-PSGD (Tang et al., 2018) and Choco-SGD (Koloskova et al., 2019)." is wrong.
>
> Second, we would like to clarify that the proposed algorithm is not aimed to train large language models, as illustrated in our experiments. It aims to train small models or fine tuning in our opinion. We believe such algorithms still have their merits since machine learning does not equal to large language model training.
>
> **Re: Assumptions**
>
> We invite the reviewer to see our response to Reviewer 7fCr on this issue. In addition, we would like to point out some mistakes in the review.
>
> First, the examples where $\beta > 1$ are wrong. For example, in [6], $\beta = \frac{1}{8}$ instead of $\frac{9}{8}$. In fact, when $\beta = 1$, it means no compression at all. Hence, any value $\beta > 1$ does not make sense. We invite the reviewer to read all the mentioned papers again.
>
> Second, our bounded compression assumption is not related to contractiveness. We believe you mean ``contraction", which has nothing to do with our assumption on the compressor. I suggest the reviewer to check the mathematical definition of contraction again carefully.
>
>
> **Re: Convergence Results**
>
> First, we don't see how the convergence bound in [3] can better characterize the impact of network topology and $\beta$ since our upper bound is also a function of these two parameters. For example, it is very clear that our convergence upper bound is monotonically increasing function of $\rho$, which is expected. We will investigate it more in the future versions of our paper.
>
> Second, we agree with the fact that we are not familiar with [3]. However, [3] came out in May, 2024 on ArXiv and the current paper has even changed the title. After a quick read of this paper, we believe the proposed MoTEFand MoTEF-VR algorithms may have a higher computation and communication cost, and may require more memories compared to our algorithm. We appreciated the reviewer pointing out this paper and we will compare our algorithm with the algorithms mentioned in this paper very carefully in the future versions of our paper.
>
> Third, the goal of our algorithm is not to improve the order of the convergence rate but to reduce the requirements (assumptions) to make the algorithm converge. For example, we do not have a high requirement for data heterogeneity and compression ratio (at least from the experiment point of view). We will make it more clear in the future versions of our paper.
>
> Fourth, it is really impolite to say our experiments are trivial. We do hope the reviewer can appreciate more on people's efforts to make something work.  We made it clear in the paper that the unbaiseness assumption is only to make the analysis more tractable. In the experiments, we showed that our algorithm can work for the compressor that does not satisfy the unbiasedness assumption. In addition, in the Appendix, we did compare our algorithm with other more recent algorithms. We will compare our algorithm with the mentioned algorithms in the future version of our paper. For the performance of CIFAR0-10, we didn't use the most sophisticated model. The model is relatively small, which is why it gives an accuracy of around $70\%$. The accuracy can be improved when using a bigger model.

---

### Official Review · Reviewer_7fCr · 2024-11-01

**Soundness:** 1
**Presentation:** 1
**Contribution:** 1
**Rating:** 1
**Confidence:** 4

**Summary:**

This paper proposes a detentralized optimization algorithm with communication compression called DEFD-PSGD, which applies the error-feedback mechanism. Under certain assumptions, DEFD-PSGD is shown to converge exactly. In numerical experiments, DEFD-PSGD is shown to perform better than CHOCO-PSGD.

**Strengths:**

1. The algorithm is simple and clear.
2. The theoretical convergence rate matches D-PSGD.
3. The proposed algorithm performs better than other baselines in the experiments.

**Weaknesses:**

1. The compared baselines are sub-optimal. By comparing with baseline algorithms DCD-PSGD and CHOCO-PSGD, which are from 2019 and earlier, it's not proper to claim that DEFD-PSGD "outperforms other state-of-the-art decentralized learning algorithms". CEDAS in 2023 has already beaten CHOCO-PSGD by a large margin.
2. The assumptions used in the theoretical analysis are too strong. Specifically:\
i) Bounded gradient divergence: To my knowledge, the aim to use error-feedback should be removing this assumption which leads to a small-heterogeneity setting. The use of this assumption seems to contradict with the aim to optimize with data heterogeneity.\
ii) Unbiased stochastic compression & Bounded compression error: Usually compressed algorithms only need to assume either unbiasedness or $\beta\le1$. Assuming both is too strong, and even contradicts with "a high degree of gradient compression".

**Questions:**

1. Could the reviewers illustrate how to design a compressor with a high degree of compression that satisfies both unbiasedness and a bounded compression error of $\beta\le 1$?
2. As in weakness 1, can the authors compare the proposed algorithm with more recent baselines, such as CEDAS?
3. As in weakness 2, why the assumptions seem contradict with high compression degree and large data heterogeneity? Please correct me if I'm wrong.

---

> ### Author Response · Authors · 2024-12-04
>
> Many thanks for your review. We appreciated some points you raised. We will revise our paper using the valuable points. Due to the extremely low and insulting score, we will withdraw our paper. However, we still would like to clarify some points that may be due to misunderstanding. From reading the reviews, we believe the reviewer's knowledge in the area of federated learning is very limited. Our responses to your comments are as follows.

---

> ### Author Response · Authors · 2024-12-04
>
> **Re: Baselines**
>
> We would like to point out that we did compare the proposed algorithm with other more recent algorithms in Appendix A.5 including AdaG-PSGD and Comp Q-SADDLe. The reason we emphasized more on DCD and CHOCO is that they are representative and most of the other improvements based on those have the same issues on data heterogeneity and/or high compression ratio. We will compare our algorithm with CEDAS in the future versions of our paper, where CEDAS was recently published (30 September 2024) in IEEE Transactions on Automatic Control.
>
> **Re: Assumptions**
>
> First, regarding the bounded gradient divergence, the reviewer claimed that "the aim to use error-feedback should be removing this assumption which leads to a small-heterogeneity setting." This is a wrong statement. The goal of error-feedback is to reduce the impact of model/gradient compression.
>
> Second, regarding the unbiased stochastic compression and bounded compression error. The reviewer claimed that "Usually compressed algorithms only need to assume either unbiasedness or $\beta < 1$. Assuming both is too strong, and even contradicts with "a high degree of gradient compression"." We agree with the reviewer that our assumptions are strong. This is mainly due to the analytical difficulty to obtain meaningful convergence upper bound. From our experimental results, it can be seen that the proposed algorithm works very well with top-k compression, which is biased. We believe the unbiasedness assumption is not needed. We will try to eliminate this assumption in the future versions of our paper. In addition, we respectively disagree that assuming both contradicts with ``a high degree of gradient compression". For example, random quantization can be both unbiased and upper bounded when the quantization level is high. For a random quantizer, it is possible that it has a fixed mean and large variance. Please also see the response for Reviewer 2PpH's related comments.

---

### Official Review · Reviewer_k2Tp · 2024-11-02

**Soundness:** 2
**Presentation:** 3
**Contribution:** 2
**Rating:** 3
**Confidence:** 4

**Summary:**

This paper proposes compressed decentralized algorithm with the aim to control the error due to data heterogeneity. The proposed algorithm DEFD-PSGD applies compressed model updates and therefore is able to perform exact model gossip under compressed communication.

**Strengths:**

DEFD-PSGD is a novel algorithm in the sense that it synchronizes model parameters by applying compressed model updates, unlike prior works [Koloskova et. al., 2019] which consider applying compressed model gossip.

**Weaknesses:**

- The convergence guarantee only holds for a restricted range of $\beta$, where $\beta$ is the relative error of contractive compressor. This is non-standard, especially along the line of work of [Koloskova et. al., 2019], [1], [2], [3] from additional references below. Can the authors explain why a restricted range of $\beta$ applies to DEFD-PSGD? I suggest the authors to expand the current analysis to the case for any $\beta \in (0, 1)$, e.g., by transferring the dependence to the parameter $\gamma$.
- Under the context of nonconvex optimization, [3] is a more accurate reference than [Koloskova et. al., 2019] for mentioning CHOCO-SGD because [3] provided the convergence of CHOCO-SGD on nonconvex objective while [Koloskova et. al., 2019] only considered convex objective.

(**Notations**)
- In equations (2), (6), (10), $\nabla f(\frac{X_t {\bf 1}_n}{n})$ should be $\\|\nabla f(\frac{X_t {\bf 1}_n}{n}) \\|^2$ instead.
- The constants $a,b,c,\mu$ are not defined in or before Theorem 1.
- In line 243 "neightbors" should be "neighbors".

**Questions:**

(**Additional References**)

Along the line of compressed decentralized algorithms for tackling data heterogeneity, I suggest the following two references that should be covered in the related works. [1] is an algorithm with large batch stochastic gradient tracking (see Theorem 4.2 of [1]) while [2] is an algorithm with constant batch stochastic gradient tracking (see Appendix B of [2]). Gradient tracking algorithms are known to converge without assuming a uniform bound on the similarity between local and global objective gradients.

(**About Comparison to CHOCO-SGD**)

The paragraph **Comparison between DEFD-PSGD and CHOCO-PSGD** is confusing and here is my point of view:

- I assume that the authors are referring CHOCO-SGD in [3] as equivalent to CHOCO-PSGD in Algorithm A.1. Otherwise, the comparison and discussion made between DEFD-PSGD and CHOCO-PSGD are not meaningful becuase DEFD-PSGD should be compared against CHOCO-SGD. However, it is not clear how to show the equivalence of Algorithm A.1 to CHOCO-SGD in [Koloskova et. al., 2019] or in [3]. So I request the authors to show a formal proof of equivalence explicitly. (A similar comparison between two error-feedback schemes is studied in [4], where Algorithm 1 in [4] has a similar error-feedback mechanism as CHOCO-SGD while Algorithm 4 in [4] has a similar error-feedback mechanism as DEFD-PSGD.)

- CHOCO-SGD is an algorithm proved to converge under any degree of data heterogeneity, for instance, their analysis in [Koloskova et. al, 2019] and [3] only assume bounded stochastic gradient and does not impose assumption on data heterogeneity. Therefore, the claim `so that when data is highly heterogeneous, the CHOCO-PSGD may diverge` in line 389 is not theoretically supported. I suggest the authors to provide more evidence for supporting the claim or revise the discussion.

- The experiment results in Figure 2 of [2] suggests that CHOCO-SGD shows slow convergence in heterogeneous distribution of MNIST. This does not match with the claim of this paper in Figure 1 which states that CHOCO-SGD cannot converge under heterogeneous data. I encourage the authors to conduct additional experiments on a similar setup of [2], to gather more insights about CHOCO-SGD and address the divergence of CHOCO-SGD with more solid evidence such as what parameter tuning is used.

- The experiment result would be more convencing if a comparison is made between DEFD-PSGD and [1], [2] on heterogeneous data setting.


(**About Data Heterogeneity**)
- Can the authors provide more insights on how DEFD-PSGD mitigates data heterogeneity through the analysis result? For instance, please explain how does the algorithm react to difference values of $\epsilon$ and whether that effectively tackles with the data heterogeneity error in terms of the convergence bound.
- It would be more convincing if the experiment result can be presented with different levels of data heterogeneity (different Dirichlet parameter $\alpha$) in the same plot and demonstrate how different algorithms react to the error of data heterogeneity.

(**About Consensus Error**)
- I suggest the authors to include a convergence bound on the consensus error in the main text and discuss how does DEFD-PSGD benefits from using an exact synchronized model gossip communication. This will provide more insight about the advantage of DEFD-PSGD, e.g., whether the dependence on $\epsilon$ is better than other algorithms (such as Theorem 4 in [Lian et. al., 2017]) in terms of consensus error bound.

[1] Haoyu Zhao, Boyue Li, Zhize Li, Peter Richtárik, and Yuejie Chi. BEER: Fast $\mathcal{O} (1/T)$ Rate for Decentralized Nonconvex Optimization with Communication Compression, 2022.

[2] Chung-Yiu Yau, Hoi-To Wai. DoCoM: Compressed Decentralized Optimization with Near-Optimal Sample Complexity, 2022.

[3] Anastasia Koloskova, Tao Lin, Sebastian U. Stich, Martin Jaggi. Decentralized Deep Learning with Arbitrary Communication Compression. 2020.

[4] Peter Richtárik, Igor Sokolov, Ilyas Fatkhullin. EF21: A New, Simpler, Theoretically Better, and Practically Faster Error Feedback, 2021.

---

> ### Author Response · Authors · 2024-12-04
>
> Many thanks for your review. We really appreciated the many points you raised. We will revise our paper using the valuable points. Due to the low score, we will withdraw our paper. However, we still would like to clarify some points that may be due to misunderstanding. Our responses to your comments are as follows.

---

> ### Author Response · Authors · 2024-12-04
>
> **Re: Issues with $\beta$.**
>
> We do agree with the reviewer that it will be ideal to extend the analysis to the entire range of $\beta \in (0,1)$. However, we respectively disagree with the reviewer for the following aspects.
>
> 1) The reviewer claimed "This is non-standard, especially along the line of work of [Koloskova et. al., 2019], [1], [2], [3] from additional references below." First, although these works do not have an explicit range on $\beta$, $\beta$ is actually a function of $\gamma'$ (see Algorithm A.1 in the Appendix), while our upper bound of $\beta$ is a function of $\gamma$. We believe our result is less constraint than the works that the reviewers mentioned. Second, in other lines of work, they do have the constraint on $\beta$. For example, in the following paper, which introduced DCD-PSGD.
>
> Tang, Hanlin, Shaoduo Gan, Ce Zhang, Tong Zhang, and Ji Liu. "Communication compression for decentralized training." Advances in Neural Information Processing Systems 31 (2018).
>
> It has a clear upper bound on $\beta$. Our bound of $\beta$ is a generalized version of it. As we discussed in our paper, our bound can be better due to the use of error feedback.
>
> 2) The reviewer mentioned that "I suggest the authors to expand the current analysis to the case for any $\beta \in (0,1)$, e.g., by transferring the dependence to the parameter ." We wanted to point out that our current upper bound of $\beta$ is already a function of $\gamma$. We can of course write an equal sign there to make it look similar to [Koloskova et. al., 2019] and "eliminate" the constraint on $\beta$ but it is fake to do so in our opinion.
>
> **Re: Issues with Algorithm A.1**
>
> Algorithm A.1 is indeed introduced in [3] (Algorithm 4 in the Appendix), where the authors (almost the same as in [Koloskova et. al., 2019]) claimed the equivalence between this algorithm and CHOCO-SGD. Hence, the comment regarding Algorithm A.1 is not valid.
>
> **Re: Regarding Algorithm 4 in [4]**
>
> The reviewer claimed that ''Algorithm 4 in [4] has a similar error-feedback mechanism as DEFD-PSGD." We respectively disagree with this claim. First, the algorithm proposed in [4] is a centralized algorithm, not for the decentralized training. Hence, it is not comparable with our algorithm. Second, [4] uses the error of the difference in gradients while our proposed algorithm uses the error of the gradient itself, where the gradient is corrected using the error that is fed back in the last iteration.
>
> **Re: CHOCO convergence under any degree of data heterogeneity**
>
> We respectively disagree with this comment and invite the reviewer to read either [Koloskova et. al., 2019] or [3] again, where the authors assumed that the stochastic gradient is bounded, which is a stronger assumption than bounded gradient divergence and implies gradient divergence. Hence, CHOCO does assume the gradient divergence is bounded so the dataset cannot be too heterogeneous.
>
> **Re: Issues with experiments**
>
> We would like to point out that [1] uses MINIST while our Figure 1 uses FashionMNIST, which is a harder dataset to learn compared to MNIST. We explained the details about our experiments and how to optimally choose all the parameters in Appendix A.1. We invite the reviewers to take a look and see if we missed anything.
>
> **Re: Data heterogeneity**
>
> This is a very good comment. We explain the the reason why the proposed algorithm can be better for high data heterogeneity in the first paragraph in page 8. In the convergence upper bound, the term in front of $\epsilon$ is not a monotonic function of $\gamma$. There exists an optimal value of $\gamma$ for the optimal data heterogeneity consideration. We will further investigate it in the future versions of our paper. For the experiments, we do have the results for different values of $\alpha$ in the appendix.
>
> **Re: Additional experiments**
>
> We would like to thank the reviewer for the comments. We will run additional experiments in the future versions of our paper.
>
> **Re: Regarding the typos**
>
> We would like to thank the reviewer to point out the typos in this paper, especially the ones in (2), (6), (10). We will revise our paper carefully and correct those typos.

---

### Note · Authors · 2024-12-04

I have read and agree with the venue's withdrawal policy on behalf of myself and my co-authors.